# Functional characterization of C21ORF2 association with the NEK1 kinase mutated in human in diseases

Mateusz Gregorczyk[1], Graziana Pastore[2,3,*], Ivan Muñoz[1,*], Thomas Carroll[1,*], Johanna Streubel[4,5,*], Meagan Munro[2,3], Pawel Lis[1], Sven Lange[1], Frederic Lamoliatte[1], Thomas Macartney[1], Rachel Toth[1], Fiona Brown[1], James Hastie[1], Gislene Pereira[4,5], Daniel Durocher[2,3], John Rouse[1]

**The NEK1 kinase controls ciliogenesis, mitosis, and DNA repair, and *NEK1* mutations cause human diseases including axial spondylometaphyseal dysplasia and amyotrophic sclerosis. *C21ORF2* mutations cause a similar pattern of human diseases, suggesting close functional links with *NEK1*. Here, we report that endogenous NEK1 and C21ORF2 form a tight complex in human cells. A C21ORF2 interaction domain "CID" at the C-terminus of NEK1 is necessary for its association with C21ORF2 in cells, and pathogenic mutations in this region disrupt the complex. AlphaFold modelling predicts an extended binding interface between a leucine-rich repeat domain in C21ORF2 and the NEK1–CID, and our model may explain why pathogenic mutations perturb the complex. We show that NEK1 mutations that inhibit kinase activity or weaken its association with C21ORF2 severely compromise ciliogenesis, and that C21ORF2, like NEK1 is required for homologous recombination. These data enhance our understanding of how the NEK1 kinase is regulated, and they shed light on NEK1–C21ORF2–associated diseases.**

## Introduction

It is over three decades since the NEK1 protein kinase was discovered, but remarkably little is known concerning the molecular mechanisms underlying the roles and regulation of this enzyme. NEK1 is one of 11 members of the NEK family of protein serine/threonine kinases related to the NIMA ("never in mitosis") kinase from *Aspergillus nidulans* (Letwin et al, 1992; Schultz & Nigg, 1993; O'Regan et al, 2007). NIMA is important for controlling mitotic entry (Morris, 1975; Oakley & Morris, 1983), and cells from NEK1-deficient mice show disordered mitoses, and errors in mitotic chromosome segregation and cytokinesis (O'Regan et al, 2007; Chen et al, 2011).

Moreover, abnormal organization of the meiosis I spindle and faulty chromosome congression has been observed in gametes from these mice (Brieno-Enriquez et al, 2017). Although NEK1 is clearly involved in mitotic function, the underlying mechanisms and the relevance of kinase activity remain to be determined.

NEK1 localises to centrosomes which, in addition to organizing the mitotic spindle, nucleate the formation of primary cilia at the cell surface (Mahjoub et al, 2005; Shalom et al, 2008). Primary cilia are single, thin projections found on the surface of most mammalian cells, predominantly on non-dividing or quiescent cells, although they can also occur in $G_1$ phase of proliferating cells (Zimmerman, 1898; Sorokin, 1968; Satir & Christensen, 2007). Cilia are tethered at their base by the mother centriole of the centrosome, from which an array of microtubules protrude, sheathed in ciliary membrane, to provide scaffolding for the axonemal projection (Kowalevsky, 1867; Sorokin, 1968). These projections are sealed off at the base from the cytosol, with cargo trafficked in and out by dedicated intra-flagellar transport proteins (Bloodgood, 1977; Kozminski et al, 1993). In this light, cilia are highly enriched for subsets of signalling proteins, acting as major hubs for Hedgehog (Shh) signalling, critical for multiple aspects of cell function (Huangfu et al, 2003; Wheway et al, 2018). Mouse and human cells lacking functional NEK1 show pronounced defects in ciliogenesis and cilium function (Evangelista et al, 2008; Shalom et al, 2008). Furthermore, Nek1-deficient mice (*kat* and *kat2j*) display symptoms typically associated with ciliopathies, diseases caused by ciliary defects, including polycystic kidney disease (Vogler et al, 1999; Upadhya et al, 2000; Mahjoub et al, 2005; Chen et al, 2014). Even though NEK1 seems to play important roles in ciliogenesis, the underlying mechanisms remain to be determined, and it is not yet clear if kinase activity is important.

NEK1-deficient mice show a range of defects beside kidney disease, including facial dysmorphism, dwarfism, cystic choroid plexus, male sterility, and anaemia (Janaswami et al, 1997; Upadhya et al, 2000). Sterility and anaemia are often associated with defects

[1]MRC Protein Phosphorylation and Ubiquitylation Unit, Wellcome Trust Biocentre, University of Dundee, Dundee, UK  [2]The Lunenfeld-Tannenbaum Research Institute, Mount Sinai Hospital, Toronto, Canada  [3]Department of Molecular Genetics, University of Toronto, Toronto, Canada  [4]German Cancer Research Centre (DKFZ), Centre for Organismal Studies, University of Heidelberg, Heidelberg, Germany  [5]DKFZ-ZMBH Alliance, Heidelberg, Germany

Correspondence: j.rouse@dundee.ac.uk
*Graziana Pastore, Ivan Muñoz, Thomas Carroll, and Johanna Streubel contributed equally to this work

in DNA repair, and in this light, NEK1 has been implicated in homologous recombination (HR), a highly conserved pathway important for repairing double-strand breaks (DSB) in proliferating cells (Spies et al, 2016). HR uses the intact chromatid present in cells in late S- and G$_2$-phases as a template for repairing DNA breaks. DSB arising in S- and G$_2$-phases are resected through the action of nucleases and helicases, resulting in formation of 3′ single-stranded (ss) overhangs (Wright et al, 2018; Bonilla et al, 2020). RAD51 coats the ssDNA to form a nucleoprotein filament which searches for sequence homology in the intact chromatid, invades the homologous duplex, and directs annealing to the homologous sequences. The invading strand is then extended using a form of non-canonical DNA synthesis (Saredi & Rouse, 2019; Bonilla et al, 2020). The final stages of HR can result in the formation of structures that entangle the two chromatids, which are enzymatically removed to complete repair (Doniger et al, 1973; Thompson et al, 1975; Lilley, 2017). Depletion of NEK1 from HeLa cells was reported to cause near-complete abrogation of DSB-induced HR; this appeared to stem from loss of NEK1-mediated phosphorylation of the RAD54 ATPase on Ser$^{572}$, which was in turn required for RAD54-mediated unloading of RAD51 from DSB repair sites (Wright & Heyer, 2014; Spies et al, 2016). However, RAD54 as a NEK1 target has been challenged recently (Ghosh et al, 2022 Preprint).

Mutations in the human NEK1 gene have been linked to several distinct disease aetiologies. For example, NEK1 mutations have been found in human patients with autosomal recessive Majewski type short-rib polydactyly syndrome, which is associated with polycystic kidneys (Thiel et al, 2011; Chen et al, 2012), lethal skeletal dysplasia, polydactyly, facial dysmorphism, drastic growth defects in utero, and microcephaly (Chen et al, 2012). NEK1 mutations have also been found in other skeletal dysplasias such as Jeune syndrome (McInerney-Leo et al, 2015) and axial spondylometaphyseal dyplasia (SMD) (Wang et al, 2017). The disease most strongly associated with NEK1 mutations is amyotrophic lateral sclerosis (ALS), a form of motor neuron disease with symptoms that do not overlap with SMD or short-rib polydactyly syndrome (Yao et al, 2021; Brenner & Freischmidt, 2022). In 2015, an exome sequencing–based report revealed heterozygous NEK1 mutations in sporadic ALS (Cirulli et al, 2015). Since then, a range of studies have validated NEK1 as an ALS gene in both sporadic ALS and familial ALS (Brenner et al, 2016; Kenna et al, 2016; Gratten et al, 2017; Nguyen et al, 2018; Shu et al, 2018; Goldstein et al, 2019; Tripolszki et al, 2019; Riva et al, 2022). The human NEK1 protein can be divided roughly into three regions. First, the kinase catalytic domain is located at the N-terminus and has sequence motifs typical of Ser/Thr kinases (Melo-Hanchuk et al, 2017). A recombinant form of the isolated kinase domain from murine Nek1 has been reported to phosphorylate Tyr residues as well as Ser/Thr residues in vitro in generic phospho-acceptor substrates (Letwin et al, 1992). In contrast, wild-type NEK1 does not phosphorylate substrates on Tyr residues in vitro (van de Kooij et al, 2019). Besides the catalytic domain, NEK1 contains a central coiled-coil region implicated in protein–protein interaction and a C-terminal acidic region of unknown function (Fig S1). The NEK1 mutations associated with ALS, SMD, and other diseases result in amino acid (aa) changes in all three regions of the protein, with little evidence for clustering. Most ALS-associated NEK1 mutations are heterozygous missense mutations that are presumed to be

dominant, generating mutant forms of NEK1 protein that interfere with the function of the product of the WT allele (Kenna et al, 2016; Goldstein et al, 2019). At present, the impact of the pathological mutations on NEK1 kinase activity is not known, and it is not understood how these mutations affect DNA repair, ciliogenesis, or mitotic progression.

NEK1 is closely linked to C21ORF2 in functional terms. First, mutations in the C21ORF2 gene phenocopy NEK1 mutations in Jeune syndrome, SMD, and ALS (van Rheenen et al, 2016; Wang et al, 2016; McInerney-Leo et al, 2017); this pattern of similarity is unique, suggesting that the functions of these two genes are intimately linked. Second, depleting C21ORF2 from hTERT–RPE1 cells caused defective ciliogenesis, similar to NEK1-deficient cells (Wheway et al, 2015). Third, at least when the proteins are over-expressed NEK1 associates with C21ORF2 (Cirulli et al, 2015; Wheway et al, 2015). These studies led us to investigate in detail the interaction of NEK1 with C21ORF2 and to test the phenotypic similarity of cells where the two genes are knocked out in an isogenic background. To this end, we set out to generate high-quality antibodies and gene knockouts (KOs) in untransformed ARPE-19 cells to assess rigorously whether NEK1 and C21ORF2 form a complex at the endogenous level in human cells; to predict the structure of the interaction interface; to characterise the complex in detail and define interaction domains; to compare directly the phenotypic defects in NEK1– and C21ORF2–KO cells; and to establish a "rescue" system to carry out structure function analysis in phenotypic analyses. Our data show that NEK1 mutations that inhibit kinase activity or weaken its association with C21ORF2 impact severely on NEK1 function.

## Results

### Generation of NEK1 and C21ORF2 antibodies and KO cell lines

To enable us to rigorously test the interaction of endogenous NEK1 and C21ORF2, we generated polyclonal antibodies in sheep. As shown in Fig 1A, affinity-purified antibodies raised against the full-length human C21ORF2 antibody recognised a band of ~28 kD in extracts of untransformed ARPE-19 retinal pigmented epithelial cells, which was reduced in intensity by a C21ORF2-specific siRNA. Similarly, NEK1 antibodies recognised a band of the expected molecular weight (~141 kD), and band intensity was reduced by an NEK1-specific siRNA (Fig 1A). C21ORF2 levels were significantly reduced in NEK1-depleted cells, consistent with a previous report (Watanabe et al, 2020). siRNA-mediated depletion of NEK1 from U2-O-S or HeLa cells also reduced C21ORF2 levels (Fig S2A and B). Intriguingly, in these cell lines (unlike ARPE-19 cells), depletion of C21ORF2 caused a slight decrease in NEK1 protein levels although the size effect was variable (Fig S2A and B). Similar data were reported in HEK293 cells (Watanabe et al, 2020).

To further test the specificity of the antibodies, the NEK1 and C21ORF2 genes were disrupted in ARPE-19 cells. CRISPR-mediated genome editing was used to disrupt NEK1 using three distinct gRNA pairs targeting exon 3 and 7. One clone was isolated which lacked any detectable NEK1 protein judged by blotting of extracts and immunoprecipitation using our NEK1 polyclonal antibodies (Fig S3A

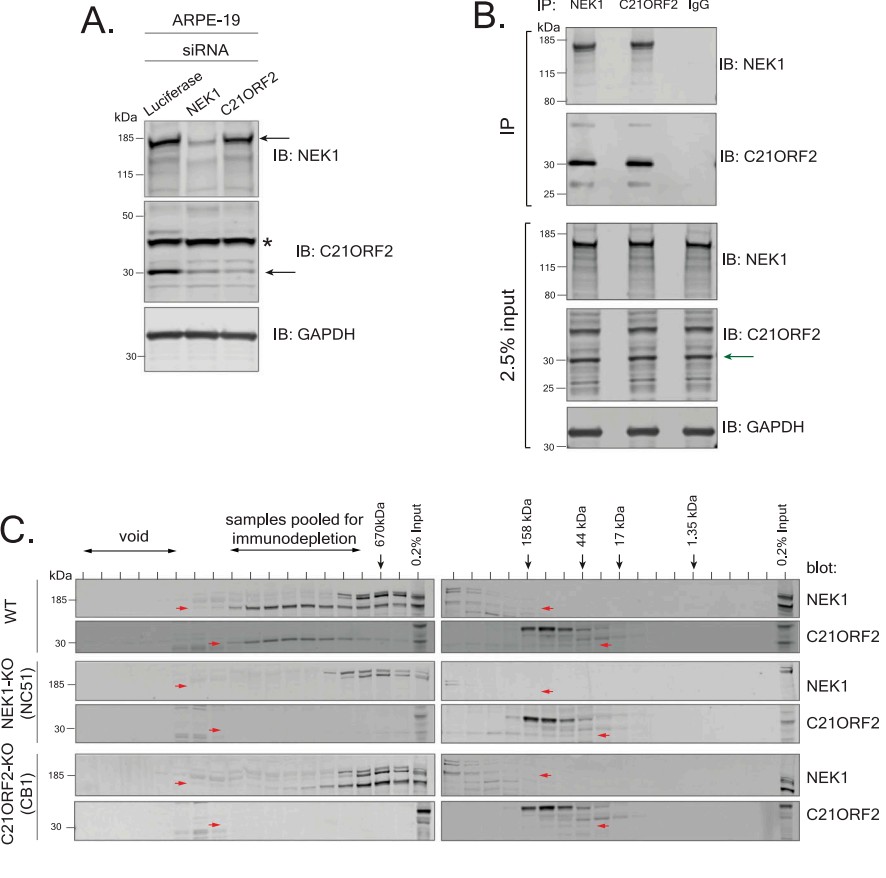

**Figure 1.   Characterization of the endogenous NEK1–C21ORF2 complex.**
**(A)** ARPE-19 cells were transfected with siRNA targeting either C21ORF2 (siRNA C21ORF2-2) or NEK1. Cell lysates were subjected to immunoblotting using in-house sheep polyclonal antibodies against C21ORF2 or NEK1 antibodies or GAPDH antibodies. Cell lysates treated with siRNA targeting luciferase served as a negative control. Asterisk denotes non-specific band; arrow denotes the specific bands. One of at least three independent experiments is shown. **(B)** ARPE-19 cells extracts were subjected to immunoprecipitation with in-house sheep antibodies against NEK1 or C21ORF2, or sheep IgG. Precipitates were subjected to SDS–PAGE and immunoblotting with the antibodies indicated, and input cell extracts were also included. Arrow denotes the C21ORF2 band. One of at least three independent experiments is shown. **(C)** Extracts from parental ARPE-19, NEK1–KO (clone NC51), and C21ORF2–KO (clone CB1) cells were subjected to size exclusion chromatography using a Superose 6 Increase 10/300 column. Alternate fractions were subjected to SDS–PAGE gel followed by immunoblotting using antibodies against NEK1 (Bethyl Laboratories) and C21ORF2. The elution positions of molecular mass markers are indicated with black arrows. Red arrows indicate bands corresponding to NEK1 or C21ORF2. One of two independent experiments is shown. **(D)** The fractions indicated in C were pooled and subjected to three rounds of immunodepletion with either in-house antibodies against NEK1 or C21ORF2 or sheep IgG antibodies. Beads and supernatants were analysed by SDS–PAGE and Western blotting the antibodies indicated. Molecular weight markers "kD" are indicated. One of two independent experiments is shown. Source data are available for this figure.

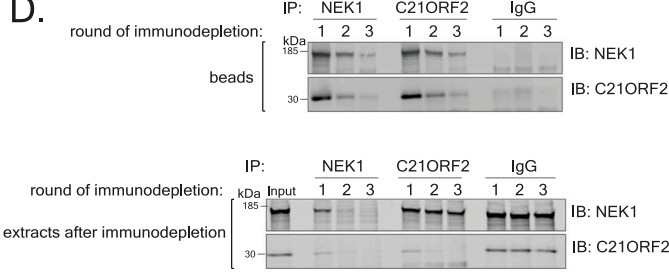

and B; clone NC51). Sequencing of the genomic DNA near the gRNA target site in clone NC51 revealed that both alleles of *NEK1* had been disrupted. As shown in Fig S3C, one allele had a large insertion at the end of exon 7, resulting in a premature stop codon, whereas the other allele had a 10 bp deletion in exon 7, generating premature stop codons at the beginning of exon 8. C21ORF2 protein levels were reduced in *NEK1* clones NC51 (Fig S3A and B), and this effect was not rescued by proteasome inhibition (MG-132), inhibition of cullin neddylation (MLN-4294), or inhibition of autophagy (bafilomycin A or ULK1 inhibitor MRT68921) (Fig S2C–F). Therefore, the fate of C21ORF2 in the absence of NEK1 is not yet clear.

Next, *C21ORF2* was disrupted using CRISPR-mediated genome editing using two pairs of sgRNAs, each targeting sequences in exon 4. Several clones lacked any detectable C21ORF2 protein judged by Western blotting of extracts (Fig S4A) and immunoprecipitation (Fig

S4B). Genotyping revealed that in clone CB1, one allele harbored two insertions in exon 4 and the other allele had simultaneous insertion and deletion, resulting in premature stop codons in both alleles (Fig S4C). Clone CB1 was taken forward for further analysis.

## Characterization of the endogenous NEK1–C21ORF2 complex

Having ascertained the specificity of our NEK1 and C21ORF2 antibodies, we used our polyclonal antibodies in immunoprecipitation experiments. As shown in Fig 1B, a strong C21ORF2 band was detected in NEK1 precipitates from extracts of ARPE-19 cells and vice versa, and similar results were obtained in the cancer cell lines HeLa, HEK293, and U2-O-S (Fig S5). The immunoprecipitation wash buffers contained 500 mM NaCl and 1% (vol/vol) Triton X-100, indicating the interaction is robust. Gel filtration of extracts of ARPE-19

cells revealed that NEK1 and C21ORF2 co-eluted in a broad peak before the 670 kD marker (Fig 1C). NEK1 eluted at a lower molecular mass upon gel filtration of extracts of C21ORF2–KO clone CB1 but not at the size expected for monomeric NEK1; C21ORF2 was only present at low levels in extracts of NEK1–KO cells and eluted at a much lower molecular mass than in parental ARPE-19 cells.

To determine the proportion of C21ORF2 that associates with NEK1, and vice versa, the gel filtration fractions from parental cells containing NEK1 and C21ORF2 were pooled and subjected to immunodepletion. As shown in Fig 1D, three rounds of depletion with NEK1 antibodies fully depleted all of the NEK1 from the pooled fractions; under these conditions, C21ORF2 was also fully depleted, indicating that all of the C21ORF2 was bound to NEK1. After three rounds of immunoprecipitation with C21ORF2 antibodies, C21ORF2 was fully depleted from the pooled fractions, but under these conditions, NEK1 was not fully depleted however, indicating that not all NEK1 is in a complex with C21ORF2 at least in ARPE-19 (Fig 1D).

Although the predicted mass of a NEK1–C21ORF2 heterodimer is ~169 kD, the gel filtration experiments showed that NEK1 and C21ORF2 co-eluted in a complex with an apparent molecular mass greater than 670 kD (Fig 1C). This discrepancy could be explained if the complex was fibrous in nature, or if multimerization occurred. Alternatively, there may be other proteins present in the complex. To find the full complement of endogenous NEK1 interactors, NEK1 was immunoprecipitated from ARPE-19 parental cells, and NEK1–KO cells were used to control for non-specific binding. Quantitative mass spectrometry revealed that the only NEK1 interactor that was at least twofold more abundant in NEK1 precipitates from parental cells compared with NEK1–KO cells was C21ORF2. Although COL8A1 and C19ORF53 were also found to differ significantly between WT and NEK1–KO cells, they were treated as contaminants because the *P*-value was close to the 0.05 cut-off (Fig 2A and Table S1). We also performed the experiment the other way round, immunoprecipitating C21ORF2 from parental ARPE-19, with C21ORF2–KO cells used as control. The only C21ORF2 interactor which was found to be at least twofold more abundant in C21ORF2 precipitates from parental cells compared with C21ORF2–KO cells was NEK1 (Fig 2B and Table S2). These data reinforce the notion that C21ORF2 is the major NEK1 interactor and argue that at least under unchallenged conditions there are no other major components of the endogenous NEK1–C21ORF2 complex. The data also strongly suggest, but do not prove, that the NEK1–C21ORF2 interaction is direct.

### Molecular determinants of the NEK1–C21ORF2 interaction

We next set out to determine the region of NEK1 interacting with C21ORF2, by investigating the ability of truncated versions of FLAG-tagged NEK1 to interact with GFP-tagged C21ORF2 in ARPE-19 cells (Fig S6A). As shown in Fig 3A, a fragment of NEK1 corresponding to the acidic domain at the C-terminus (aa 760–1,286) coprecipitated with C21ORF2, similar to full-length NEK1, whereas fragments corresponding to the kinase domain plus basic domain (aa 1–379), or the coiled-coil domain (aa 379–760) did not. We tried to narrow down the C21ORF2-interacting domain further by making more NEK1 deletion constructs (Fig S6A). These efforts showed that a small C-terminal fragment of NEK1 corresponding to aa 1,160–1,286 coprecipitated with C21ORF2, whereas deletion of this region prevented NEK1 from interacting with C21ORF2 (Fig 3B). Thus, a small domain

between residues 1,160 and 1,286 within the acidic region of NEK1 is necessary and sufficient to interact with C21ORF2—we refer to this region as the NEK1-CID "C21ORF2-interacting domain."

A small number of pathogenic mutations affect amino acids within or near the NEK1–CID defined above, raising the possibility that these mutations could disrupt the NEK1–C21ORF2 complex (Figs S1 and S6A). These include a mutation found in both ALS and SMD which truncates NEK1 at residue S1036 (S1036*) (Kenna et al, 2016; Wang et al, 2017) and the D1277A mutation found in SMD (Wang et al, 2017) (Fig S6A). Recent gene-burden analyses have revealed that R261H mutation in *NEK1* is the most prevalent *NEK1* ALS-associated mutation in the European patient cohort, and we also engineered this mutation in *NEK1* (Kenna et al, 2016) (Fig S6A). To test the impact of these mutations, ARPE-19 cells were transiently co-transfected with expression plasmids encoding N-terminal–tagged 3HA–C21ORF2 and either WT or the NEK1 pathogenic variants described above (tagged with 3xFLAG on the N-terminus). As shown in Fig 3C, 3xHA–C21ORF2 was detected in 3xFLAG–NEK1 precipitates, and the R261H NEK1 variant had no detectable impact on binding to C21ORF2. However, the S1036* and D1277A NEK1 variants showed a dramatic reduction in their association with C21ORF2 (Fig 3C). We also noted that a "kinase-dead" (KD) *NEK1* mutant bearing a substitution in the ATP-binding pocket (D146A) was indistinguishable from WT NEK1, suggesting that NEK1 kinase activity does not influence its binding to C21ORF2 (Fig 3C). Intriguingly, overexpression of NEK1 WT induced an electrophoretic mobility shift in C21ORF2 that was not seen with the D146A NEK1 kinase–inactive mutant. This suggests C21ORF2 is phosphorylated by NEK1, consistent with a previous report (Watanabe et al, 2020).

The R73P and L224P aa substitutions in C21ORF2 found in ALS (van Rheenen et al, 2016) and Jeune syndrome, respectively, might abolish interaction with NEK1 (Fig S6B) (Wheway et al, 2015). To test this possibility, ARPE-19 cells were co-transfected with plasmids encoding FLAG–NEK1 and either 3xHA–C21ORF2 (WT or mutants). In addition, a C21ORF2 T150I mutant, recognised as the most prevalent C21ORF2 mutation in the European and US cohorts of ALS patients, was included (van Rheenen et al, 2016) (Fig S6B). As shown in Fig S6C, FLAG–NEK1 was detected in HA–C21ORF2 precipitates, and the R73P and L224P C21ORF2 substitutions weakened (but did not abolish) the interaction with NEK1. In contrast, the C21ORF2 T150I mutant did not impact binding to NEK1. Taken together, the data above indicate that the NEK1–C21ORF2 association is negatively impacted by NEK1 pathogenic mutations S1036* and D1277A, and by the C21ORF2 R73P and L224P mutations.

### AlphaFold-based structural modelling of the NEK1–C21ORF2 interaction interface

We next used the ColabFold notebook (Mirdita et al, 2022) to run the AlphaFold structure prediction (Jumper et al, 2021) of the NEK1–C21ORF2 complex, using full-length C21ORF2 and the NEK1–CID (aa 1,160–1,286) as input sequences. AMBER structure relaxation was used to ensure appropriate orientation of the side chains to avoid steric clashes. Five models of a NEK1–C21ORF2 complex were generated that were ranked from higher to lower confidence based on inter-chain predicted alignment error (inter-PAE) values (Supplemental Data 1) (Mirdita et al, 2022). All five models predicted a highly consistent binding interface between a region in the

A.

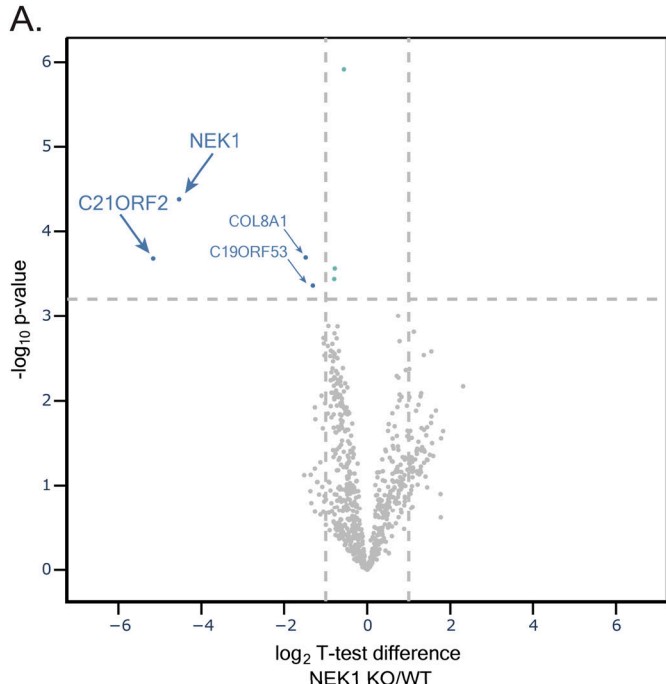

B.

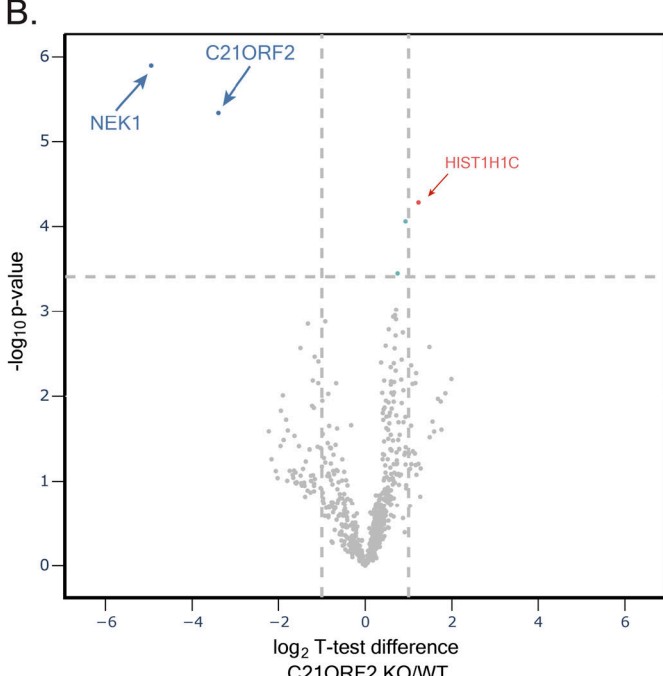

**Figure 2. Mass spectrometric analysis of the NEK1–C21ORF2 complex in ARPE-19 cells.**

**(A)** Lysates of ARPE-19 parental cells and NEK1–KO cells were subjected to immunoprecipitation with in-house sheep anti-NEK1 antibodies (five biological replicates per cell line). Proteins were eluted from beads, loaded on S-Trap columns, and after trypsinization, TMT-labelled samples were pooled and injected on an UltiMate 3000 RSLCnano system coupled to an Orbitrap Fusion Lumos Tribrid Mass Spectrometer. A volcano plot representing NEK1 interactors is shown. The horizontal cut-off line represents a P-value of 0.05, and the vertical cut-off lines represent a $\log_2$ fold change above which peptides were considered to differ significantly in abundance between ARPE-19 WT and NEK1–KO cells.

N-terminal half of C21ORF2 (aa 1–138) and a stretch of the NEK1–CID between residues 1,208 and 1,282 with high pLDDT and global PAE confidence scores for interacting regions (Figs 4A; Fig S7A includes regions predicted to be disordered). This region of C21ORF2 appears to form a leucine-rich repeat (LRR) domain where four parallel beta strands contact a series of alpha helices in the NEK1–CID located between residues 1,208 and 1,286. Asp1277 of NEK1 is predicted to form a hydrogen bond with Asn24 of C21ORF2, providing an explanation for why substitution of Asp1277 for Ala (D1277A) perturbs the association with C21ORF2 (Fig 4B). Arg73 of C21ORF2 lies in a loop connecting two LRR repeats in the NEK1 interaction interface (Fig 4B). The kink introduced by the pathogenic R73P substitution likely disrupts the LRR domain–mediating interaction with NEK1, which could explain why this substitution weakens association of C21ORF2 with NEK1 (Fig 4B).

The predicted interface between the N-terminal half of C21ORF2 (aa 1–138) and the NEK1–CID between residues 1,208 and 1,286 described above represents an extensive binding surface, with multiple contacts involving mostly electrostatic interactions and hydrogen bonds (Fig 4C). A series of positively charged residues in C21ORF2 (Lys13, Arg21, Lys22, Arg29, Arg70, Lys120) are predicted to form salt bridges with a series of negatively charged residues within NEK1–CID (Glu1215, Glu1216, Glu1235, Glu1282, Asp1283) on a surface resembling a molecular Velcro (Fig 4C). To test this binding model, we assessed the impact of C21ORF2 mutations (K13E+R21E+K22E; "3KRE" mutant) and NEK1 mutations (E1215K+E1216K+E1235K: "3EK" mutant) on partner binding. As shown in Fig 4D, the NEK1–3EK and C21ORF2–3KRE proteins show a pronounced reduction in ability to coprecipitate with partner protein, consistent with the AlphaFold model above.

The top-ranked AlphaFold model also predicted a second binding interface involving the NEK1–CID helical core and a region at the C-terminus of C21ORF2 between residues 210 and 256 albeit with low confidence (Fig 4A and B). Here, two alpha helices of C21ORF2 are predicted to bind to the backside of the NEK1–CID. It is noteworthy that the pathogenic aa substitution L224P found in Jeune syndrome locates to the first of the two helices in the predicted interface, which may account for why this substitution weakens C21ORF2 association with NEK1 (Fig 4B). However, further structural analyses are needed to determine if this second interface is valid.

### A complementation system for structure–function–based phenotypic analyses

In the next phase of the study, we set out to test if C21ORF2–KO cells show functional defects similar to NEK1–KO cells, and to test the functional impact of the pathogenic mutations which affect the interaction of NEK1 and C21ORF2. Because the functional relevance of NEK1 kinase activity is unclear, we also wished to test the impact of a kinase-dead mutant. To enable these experiments, we established a system for re-expressing NEK1 or C21ORF2 in the respective KO cell lines, which would not rely on protein overexpression. NEK1–KO cells were transduced with lentiviruses expressing NEK1

**(B)** Same as in A except extracts of ARPE-19 parental cells and C21ORF2–KO cells were subjected to immunoprecipitation with anti-C21ORF2 antibodies.

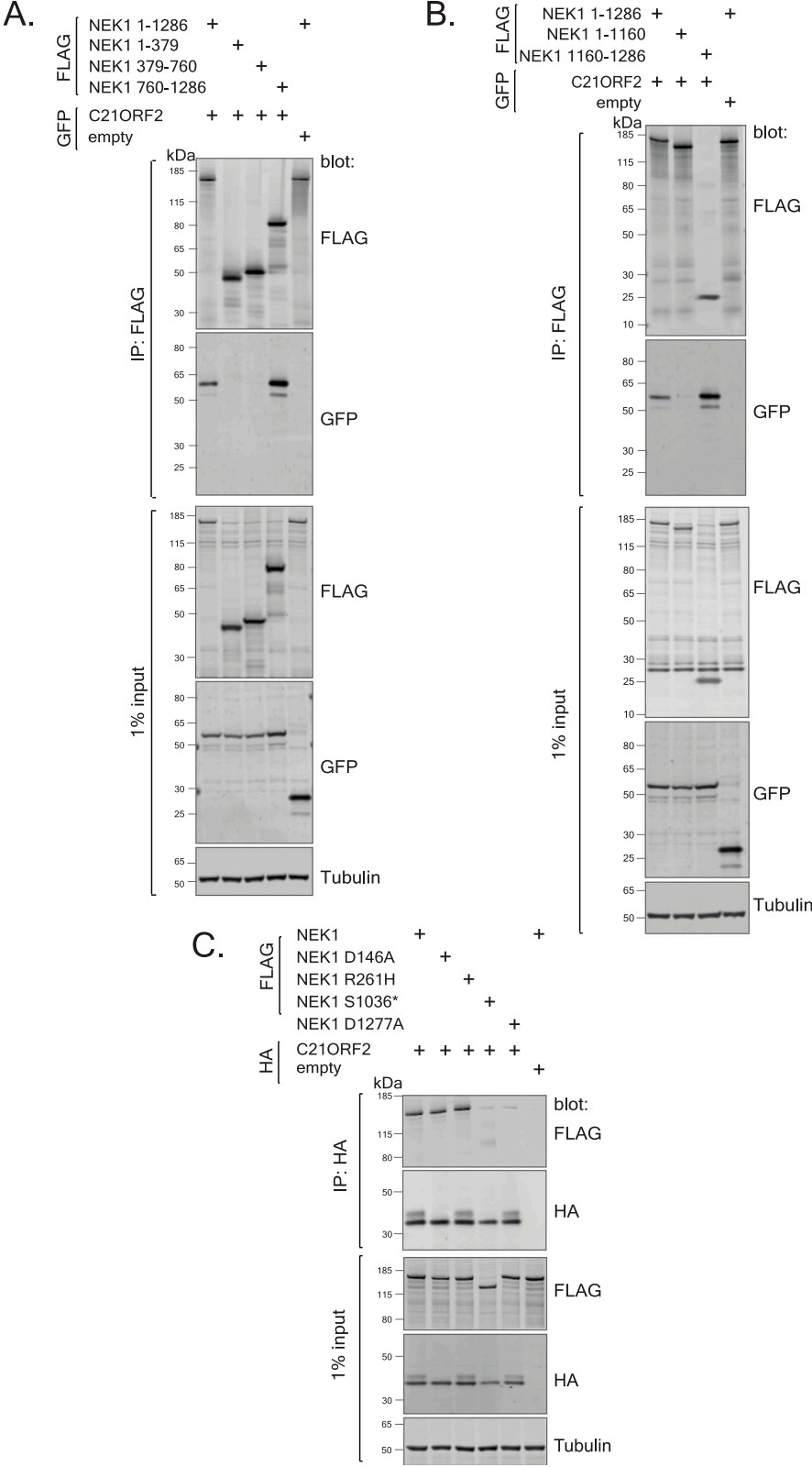

**Figure 3. Molecular determinants of the NEK1–C21ORF2 interaction.**
**(A, B)** Lysates of ARPE-19 transiently co-transfected with plasmids encoding C21ORF2 (tagged with GFP on the N-terminus) and either full length or truncated forms of NEK1 (tagged with a 3xFLAG tag on the N-terminus) were subjected to immunoprecipitation with anti-FLAG antibodies. Precipitates (and input cell extracts) were subjected to SDS–PAGE and immunoblotting with the indicated antibodies. One of two independent experiments is shown in each case. **(C)** Same as (A, B), except that ARPE-19 cells were co-transfected with cDNA encoding for C21ORF2 (tagged with 3xHA tag on the N-terminus) and WT or mutated versions of NEK1 (tagged with a 3xFLAG tag on the N-terminus). Molecular weight markers "kD" are indicated. One of two independent experiments is shown.
Source data are available for this figure.

under the control of the CMV promoter which allowed expression at close to endogenous levels (Fig S7B). A range of NEK1 mutants were also expressed: kinase-dead (KD) NEK1 (D146A), S1036*, and D1277A. Consistent with transient overexpression-based experiments shown in Fig 3, the NEK1 D1277A mutant was unable to interact with C21ORF2 (Fig S7B). We noticed that both

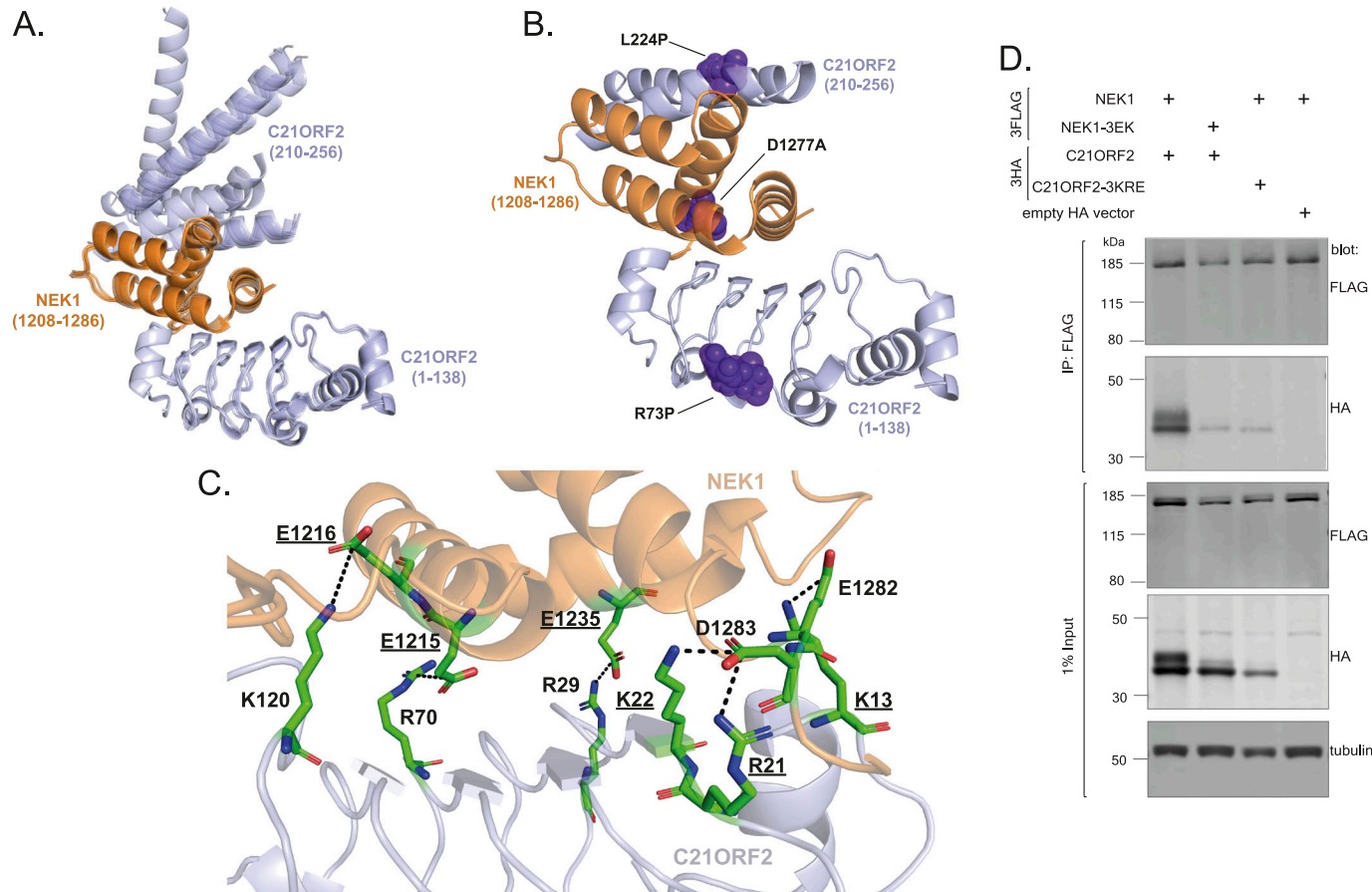

**Figure 4.  AlphaFold structural modelling of the NEK1–C21ORF2 interaction interface.**
**(A)** Overlay of the 5 models of the NEK1–C21ORF2 interaction interface generated by AlphaFold, with full-length C21ORF2 and the NEK1–CID (aa 1,160–1,286) as the input sequences. The regions of C21ORF2 corresponding to residues 1–138 and residues 210–256 involved in NEK1 interaction are shown in light blue, whereas NEK1 residues 1,208–1,286 are shown in orange. **(B)** Amino acid substitutions encoded by pathological mutations in C21ORF2 (Arg73Pro [R73P], Leu224Pro [L224P]), and NEK1 (Asp1277Ala [D1277A]), which disrupt the complex, map to the interaction interface predicted by the top-ranked AlphaFold model. The substitutions are marked in violet. The regions of C21ORF2 corresponding to residues 1–138 and residues 210–256 involved in NEK1 interaction are shown in light blue, whereas NEK1 residues 1208–1286 are shown in orange. **(C)** The binding interface between residues 1–138 in C21ORF2 (in light blue) and NEK1 (in orange) in the top-ranked model highlighting amino acids that may mediate protein–protein interactions. Charged residues forming salt bridges between the proteins are marked (positively charged in blue, negatively charged in red). Residues selected for mutagenesis are underlined. **(D)** Lysates of ARPE-19 transiently co-transfected with plasmids encoding for C21ORF2 or C21ORF2-3KRE (tagged with 3XHA on the N-terminus) and NEK1 or NEK1-3EK (tagged with a 3xFLAG tag on the N-terminus). Precipitates and extracts were subjected Western blotting with the antibodies indicated. One of two independent experiments is shown.
Source data are available for this figure.

NEK1–WT and NEK1–KD restored C21ORF2 expression level back to normal in NEK1–KO cells (Fig S7B). However, the NEK1 mutants which weakened association with C21ORF2 did not rescue C21ORF2 expression, consistent with the possibility that complex formation regulates C21ORF2 stability (Fig S7B). We also transduced C21ORF2–KO cells with lentiviruses expressing C21ORF2 under the control of the UbC promoter which allowed expression at close to endogenous levels (Fig S7C). We also introduced two pathogenic C21ORF2 mutants—L73P and L224P—found in ALS and Jeune syndrome, respectively, which show reduced association with NEK1 (Fig S7).

## Characterization of the role of NEK1 kinase and C21ORF2 in ciliogenesis

NEK1 has been implicated in a range of cellular functions, and we wondered if mutations in NEK1 that affect the interaction with C21ORF2

also affect these functions, starting with the control of primary cilia. We first tested localization of NEK1 or C21ORF2 to centrosomes and cilia. As shown in Fig 5A and B, both NEK1 and C21ORF2 co-localize with γ-tubulin at centrosomes in proliferating ARPE-19 cells, and to the ciliary base in serum-starved cells (Fig 5C), similar to previous reports (Mahjoub et al, 2005; Lai et al, 2011). To examine ciliogenesis, ARPE-19 cells were serum-starved for 48 h to initiate exit to $G_0$, and primary cilia were analysed by immunofluorescence using antibodies against ARL13B, an ARF-family small G protein highly enriched at the ciliary membrane, and pericentrin to mark centrosomes. As shown in Fig 5D and E, the proportion of NEK1–KO and C21ORF2–KO cells bearing primary cilia was reduced dramatically compared with parental cells. We noticed in some cells that an ARL13B dot co-localized with a centrosome, suggesting that establishment of the ciliary membrane may have started but axonemal microtubule extension had failed (Fig 5D).

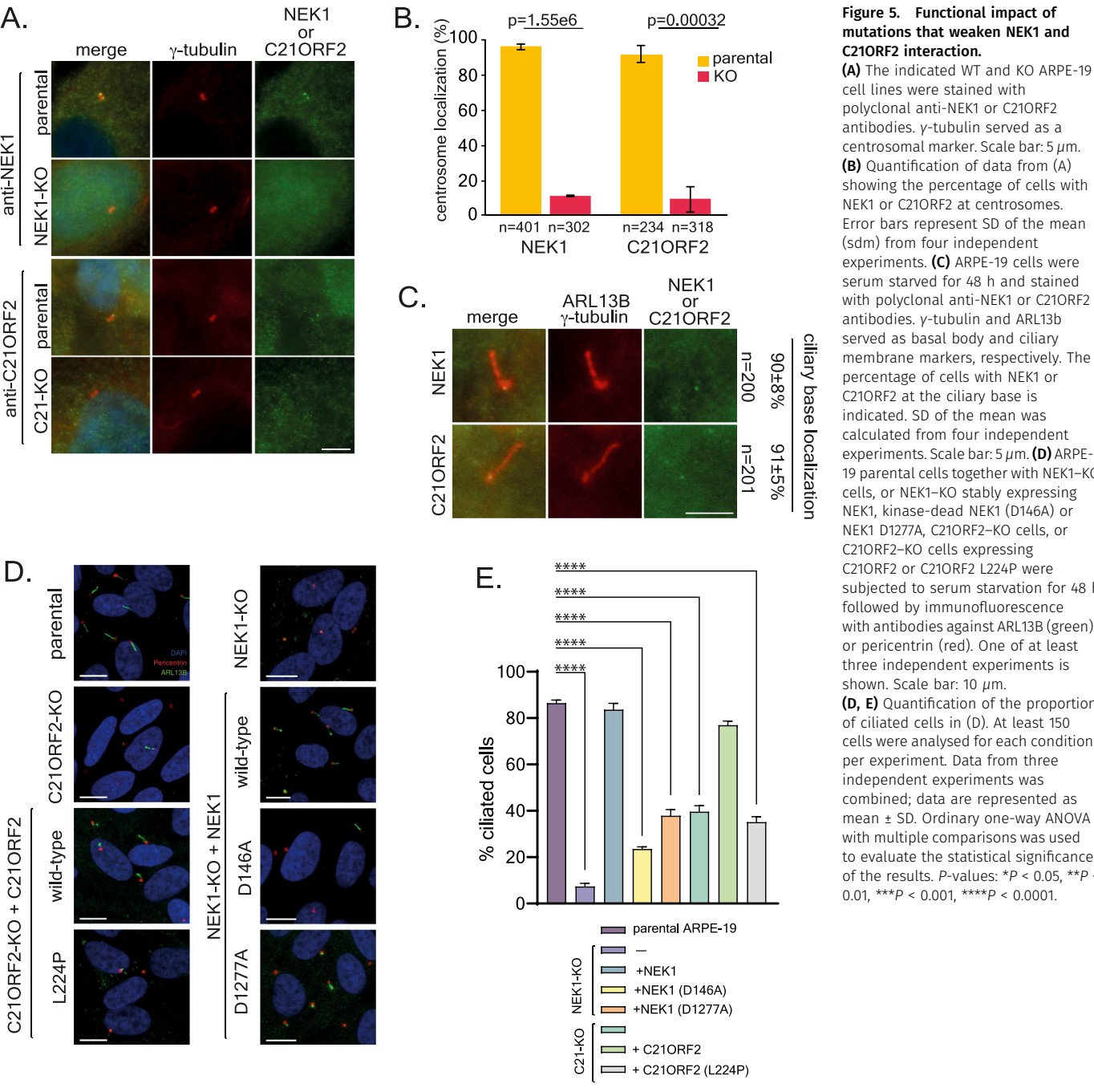

**Figure 5. Functional impact of mutations that weaken NEK1 and C21ORF2 interaction.**
**(A)** The indicated WT and KO ARPE-19 cell lines were stained with polyclonal anti-NEK1 or C21ORF2 antibodies. γ-tubulin served as a centrosomal marker. Scale bar: 5 μm. **(B)** Quantification of data from (A) showing the percentage of cells with NEK1 or C21ORF2 at centrosomes. Error bars represent SD of the mean (sdm) from four independent experiments. **(C)** ARPE-19 cells were serum starved for 48 h and stained with polyclonal anti-NEK1 or C21ORF2 antibodies. γ-tubulin and ARL13b served as basal body and ciliary membrane markers, respectively. The percentage of cells with NEK1 or C21ORF2 at the ciliary base is indicated. SD of the mean was calculated from four independent experiments. Scale bar: 5 μm. **(D)** ARPE-19 parental cells together with NEK1–KO cells, or NEK1–KO stably expressing NEK1, kinase-dead NEK1 (D146A) or NEK1 D1277A, C21ORF2–KO cells, or C21ORF2–KO cells expressing C21ORF2 or C21ORF2 L224P were subjected to serum starvation for 48 h followed by immunofluorescence with antibodies against ARL13B (green) or pericentrin (red). One of at least three independent experiments is shown. Scale bar: 10 μm.
**(D, E)** Quantification of the proportion of ciliated cells in (D). At least 150 cells were analysed for each condition per experiment. Data from three independent experiments was combined; data are represented as mean ± SD. Ordinary one-way ANOVA with multiple comparisons was used to evaluate the statistical significance of the results. *P*-values: *P < 0.05, **P < 0.01, ***P < 0.001, ****P < 0.0001.

Ciliogenesis was restored to normal by expression of NEK1 in NEK1–KO cells but NEK1–KO cells expressing the kinase-dead NEK1 D146A mutant or the D1277A mutant that shows reduced association with C21ORF2 displayed major defects in ciliogenesis (Fig 5D and E). Expression of C21ORF2 in C21ORF2–KO cells rescued the ciliogenesis defects but the C21ORF L227P mutant that shows reduced association with NEK1 was unable to affect full rescue (Fig 5D and E). Therefore, primary ciliogenesis is inhibited by loss of NEK1 kinase activity and by mutations that perturb the association of NEK1 and C21ORF2.

## C21ORF2-like NEK1 is required for homologous recombination

We next tested if C21ORF2 is, like NEK1, required for HR. We first used the "traffic light reporter" (TLR) system integrated into the genome of U2-O-S cells (Certo et al, 2011) (Fig S8A). TLR cells were transfected with siRNA targeting NEK1 or C21ORF2, with BRCA1 siRNA or non-targeting siRNA serving as positive and negative controls, respectively. As shown in Fig 6A, around 1.1% of cells transfected with siCTRL underwent gene conversion after DSB induction by overexpression of the I-SceI endonuclease. Depletion of NEK1 and

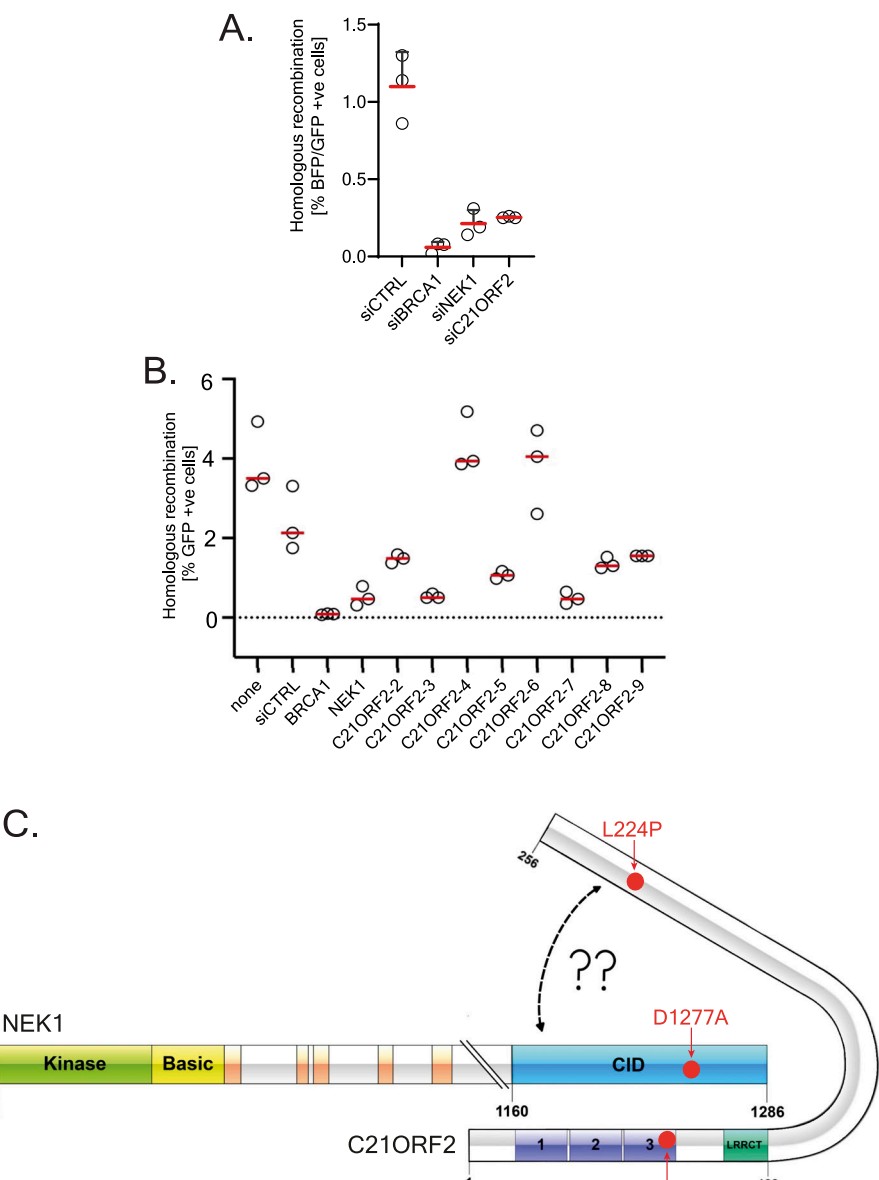

**Figure 6. C21ORF2-like NEK1 is required for homologous recombination.**
**(A)** U2-O-S TLR cells were transfected with the siRNAs indicated (C21ORF2-2 siRNA was used), and 48 h post-transfection, cells were nucleofected with a bicistronic vector–encoding GFP template and I-SceI nuclease. 24 h post plasmid transfection, cells were harvested and analysed by flow cytometry for double-positive BFP/GFP signals. Results from three independent experiments are shown. **(B)** DR-GFP U-2-O-S cells were transfected with siRNA targeting either BRCA1, NEK1, or C21ORF2 or a non-targeting siRNA (siCTRL). 24 h later, cells were nucleofected with a plasmid-encoding I-SceI nuclease. 48 h post plasmid transfection, cells were harvested and analysed by flow cytometry for GFP-positive cells. Results from three independent experiments are shown. **(C)** Schematic diagram of the NEK1–C21ORF2 complex. Globular domains in each protein are indicated. The NEK1–CID is highlighted in blue. AlphaFold modelling predicts with high confidence an interface between the N-terminal LRR–containing domain of C21ORF2 and the NEK1–CID. The top-ranked model predicts a second interface between the C21ORF2 C-terminal helices and the backside of the NEK1–CID, but the confidence of this prediction is low and needs to validated.
Source data are available for this figure.

C21ORF2 resulted in a dramatic reduction in HR efficiency, similar in effect size to depletion of BRCA1. We also used the well-characterised DR-GFP reporter system integrated into a U2-O-S genome which contains two *GFP* genes in a tandem arrangement (Fig S8B) (Pierce et al, 1999). DR-GFP cells transfected with siRNA-targeting NEK1 or with eight different C21ORF2 siRNAs were transfected later with a plasmid-encoding I-SceI and analysed by flow cytometry 48 h later. As shown in Fig 6B, gene conversion at the I-SceI–induced DSB was observed in around 4% of cells transfected with I-SceI. Depletion of BRCA1 resulted in a dramatic reduction in HR efficiency, and knockdown of NEK1 led to a comparable reduction (Fig 6B). Six of the eight C21ORF2 siRNAs we tested caused a major reduction in HR. As HR only operates in S/G2 phases, we tested if the HR defects above could reflect altered cell cycle distribution. Depleting NEK1 caused a reduction in the relative proportion of S-phase cells, whereas depleting

C21ORF2 had little effect (Fig S8C and D). Given that the reduction in HR level after depletion of either NEK1 or C21ORF2 is similar in effect size, its unlikely cell cycle distribution is responsible for the HR defect. Taken together, we conclude that cells deficient in NEK1 or C21ORF2 have a major defect in HR.

## Discussion

The NEK1–C21ORF2 interaction was first identified in experiments where over-expressed forms of NEK1 or C21ORF2 were subjected to IP/mass spectrometry analyses (Cirulli et al, 2015; Wheway et al, 2015). Here, we have presented biochemical characterization of the endogenous NEK1–C21ORF2 complex in human cells (Fig 6C). In

ARPE-19 cells, all of the C21ORF2 protein appears to be bound to NEK1, and there is a pool of NEK1 that is not bound to C21ORF2. It is not yet clear if the two partner proteins interact directly, but several observations argue in favour of this possibility. First, IP/MS analyses indicated NEK1 is the only major C21ORF2-interacting protein in unperturbed cells, and vice versa, and no other complex components were detected (Fig 2). In this light, the NEK1 has been reported to interact with other proteins after exposure of cells to cisplatin (Melo-Hanchuk et al, 2017), and C21ORF2 forms a complex with SPATA7 (Wheway et al, 2015). However, these studies involved bait over-expression, whereas our study analysed the endogenous proteins. If the interaction of endogenous NEK1 with C21ORF2 was indirect, we would expect other proteins in the complex. Second, AlphaFold modelling provided high-confidence molecular view of a direct interaction between NEK1 and C21ORF2 which was supported experimentally. Third, mutations in either partner that reduce the association perturb ciliogenesis. These observations suggest but do not prove that the interaction is direct. Demonstrating that recombinant proteins interact in vitro with a reasonable binding constant would be informative, but so far we have been unable to express recombinant C21ORF2 in soluble form. Reconstituting the NEK1–C21ORF2 complex in vitro and investigating the impact of the mutations we have described in the study that reducing the interaction in cells will be an important area of investigation.

We identified a small domain between aa 1,160 and 1,286 within the acidic C-terminal region of NEK1 which we termed the CID, that is necessary and sufficient for association with C21ORF2 (Fig 6C). The AlphaFold modelling predicted an extensive interaction interface between the N-terminal LRR region of C21ORF2 and the NEK1–CID. This interface has an extended strip of positively charged residues lining the C21ORF2 side and a strip of negatively charged residues on NEK1 essential complex formation (Fig 4C). The model may explain why the ALS-associated D1277A NEK1 variant shows reduced association with C21ORF2 and defects in ciliogenesis (Fig 6C). A second interface between the C21ORF2 C-terminal helices and the backside of the CID was predicted with low confidence (Fig 6C); if valid, this model could potentially explain the pathogenicity of the L73P and L224P aa substitutions in C21ORF2 found in ciliopathies (Wheway et al, 2015). It will be important to validate the predictions made here, by solving the structure of the NEK1 C-terminus bound to C21ORF2, and it will be interesting to determine if the interaction in cells is regulated.

C21ORF2 was identified in a genome-wide screen for factors affecting ciliogenesis, and the same study identified C21ORF2 aa substitutions associated with the Jeune syndrome, a known ciliopathy (Wheway et al, 2015). Here, we show that ARPE-19 cells lacking C21ORF2, like cells lacking NEK1, also show major defects in ciliogenesis. Furthermore, mutations in NEK1 or C21ORF2 that weaken their association are unable to fully rescue the ciliogenesis defect in the respective KO cells (Fig 5). These data suggest that the interaction of the two proteins is critical for driving ciliogenesis, but more work is needed to prove this theory. We also found that the kinase activity of NEK1 is essential for ciliogenesis (Fig 5E), but at present the targets of NEK1 required for ciliogenesis are unknown. No systematic phosphoproteomic screen for NEK1 targets has been reported. Therefore, identifying the substrates of NEK1 that are critical for ciliogenesis and testing if C21ORF2 impacts on NEK1 substrate phosphorylation will be important going forward. In this light, C21ORF2 appears to be a substrate of NEK1. Overexpression of

NEK1 but not the D146A kinase-dead mutant induced an electrophoretic mobility shift in co-expressed C21ORF2 (Fig 3C). This suggests C21ORF2 is a NEK1 target, consistent with a previous report (Watanabe et al, 2020). The same study suggested that NEK1-mediated phosphorylation is necessary for C21ORF2 stability in overexpression-based experiments (Watanabe et al, 2020), but in our hands, the reduced levels of C21ORF2 protein in NEK1–KO cells were restored by both WT and kinase-dead NEK1 (Fig S7B). It will be important to map the phosphorylation sites in endogenous C21ORF2 and to investigate their functional significance. Identifying the full complement of nuclear NEK1 targets will also be important given that we observed a major decrease in HR efficiency in cells depleted of NEK1, and in cells depleted of C21ORF2 (Fig 6A and B). It is interesting to note that the symptoms associated with SMD, for example, caused by NEK1 or C21ORF2 mutations, are somewhat reminiscent of the SPONASTRIME syndrome caused by mutations in TONSL that weaken HR (Burrage et al, 2019; Chang et al, 2019). Therefore, it is possible that SMD symptoms are caused by defective HR, and this idea will be important to test.

Mutations in *C21ORF2* and *NEK1* have been associated with an overlapping set of distinct disease aetiologies, although in some cases it is not yet clear if the mutations are causal. The strongest link is to ALS, replicated across several independent studies, with *NEK1* now regarded as a bona fide ALS gene. However, it is not yet clear how the disease-associated mutations in NEK1–C21ORF2 affect the activity of the complex in cells at the molecular level. Do *NEK1* and *C21ORF2* mutations found in different diseases affect different aspects of NEK1–C21ORF2 function—for example—ciliogenesis versus DNA repair? Do *NEK1* and *C21ORF2* mutations found in different diseases affect different subsets of NEK1 kinase targets? Again, this will require identification and robust validation of NEK1 targets. Understanding the molecular defects associated with deficiency in the NEK1–C21ORF2 complex may open avenues for new treatment for diseases caused by mutations in *NEK1* or *C21ORF2*.

NEK1, like NIMA, is required for mitotic function and spindle integrity in mouse cells (Chen et al, 2011; Brieno-Enriquez et al, 2017) but it is not yet clear if this role requires association with C21ORF2. To rigorously test their roles in mitosis, we may need a way to switch off NEK1 and C21ORF2 conditionally, for example, using degron tags. It is tempting to speculate that the role of NEK1 in mitosis involves the phospho-dependent control of centrosomes and/or microtubules. The mitotic substrates could also be relevant to ciliogenesis which, like spindle formation, requires plus-end microtubule extension from the centrosome. It is even possible that the role of NEK1–C21ORF2 in HR could be explained by microtubule regulation given reported links between the cytoskeleton, DSB mobility, and DNA repair (Lottersberger et al, 2015). This will require further work, and finding the substrates of NEK1–C21ORF2 in cells is a key priority that could provide valuable clues.

# Materials and Methods

### Reagents

All plasmids used in this study are listed in Table 1, and data sheets can be found at https://mrcppureagents.dundee.ac.uk/. Oligos corresponding to sgRNA sequences are listed in Table 2, and all

**Table 1. Plasmids used in this work.**

| Protein expressed | Plasmid | Catalogue number |
|---|---|---|
| 3xFLAG–NEK1 | pcDNA5 FRT/TO-CMV | DU58176 |
| 3xFLAG–NEK1 D146A (KD) | pcDNA5 FRT/TO-CMV | DU63607 |
| 3xFLAG–NEK1 R261H | pcDNA5 FRT/TO-CMV | DU67947 |
| 3xFLAG–NEK1 S1036* (nt change C3107G) | pcDNA5 FRT/TO-CMV | DU63628 |
| 3xFLAG–NEK1 D1277A (nt change A3830C) | pcDNA5 FRT/TO-CMV | DU63629 |
| 3xFLAG–NEK1 1–379 | pcDNA5 FRT/TO-CMV | DU67494 |
| 3xFLAG–NEK1 379–760 | pcDNA5 FRT/TO-CMV | DU67594 |
| 3xFLAG–NEK1 760–1,286 | pcDNA5 FRT/TO-CMV | DU67595 |
| 3xFLAG–NEK1 1–1,160 | pcDNA5 FRT/TO-CMV | DU67593 |
| 3xFLAG–NEK1 1,160–1,286 | pcDNA5 FRT/TO-CMV | DU68531 |
| 3xHA–C21ORF2 | pcDNA5 FRT/TO-UbC | DU70174 |
| 3xHA–C21ORF2 R73P | pcDNA5 FRT/TO-UbC | DU70225 |
| 3xHA–C21ORF2 T150I | pcDNA5 FRT/TO-UbC | DU70230 |
| 3xHA–C21ORF2 L224P | pcDNA5 FRT/TO-UbC | DU70247 |
| 3xHA–empty | pcDNA5 FRT/TO-UbC | DU70173 |
| GFP–NEK1 | pcDNA5 FRT/TO-CMV | DU58446 |
| GFP–C21ORF2 | pcDNA5 FRT/TO-CMV | DU67626 |
| GFP–empty | pcDNA5 FRT/TO-CMV | DU41455 |
| NEK1 | pLV(exp)-puro-CMV | DU70683 |
| NEK1 D146A (KD) | pLV(exp)-puro-CMV | DU61644 |
| NEK1 S1036* | pLV(exp)-puro-CMV | DU61594 |
| NEK1 D1277A | pLV(exp)-puro-CMV | DU61595 |
| CMV empty | pLV(exp)-puro-CMV | DU70925 |
| C21ORF2 | pLV(exp)-puro-UbC | DU70547 |
| C21ORF2 R73P | pLV(exp)-puro-UbC | DU70719 |
| C21ORF2 L224P | pLV(exp)-puro-UbC | DU70720 |
| UbC-empty | pLV(exp)-puro-UbC | DU70930 |
| NEK1–KO NA sense | pBabeD P U6 | DU57080 |
| NEK1–KO NA antisense | pX335 | DU57088 |
| NEK1–KO NB sense | pBabeD P U6 | DU57081 |
| NEK1–KO NB antisense | pX335 | DU57089 |
| NEK1–KO NC sense | pBabeD P U6 | DU57080 |
| NEK1–KO NC antisense | pX335 | DU57088 |
| C21ORF2–KO CA sense | pBabeD P U6 | DU64911 |
| C21ORF2–KO CA antisense | pX335 | DU64913 |
| C21ORF2–KO CB sense | pBabeD P U6 | DU64912 |
| C21ORF2–KO CB antisense | pX335 | DU64914 |

siRNA sequences used in this study are listed in Table 3. All antibodies used are shown in Table 4.

## Antibody production

Polyclonal NEK1 and C21ORF2 antibodies were raised in sheep by MRC-PPU Reagents and Services (University of Dundee) and purified against the relevant antigen (after depleting antibodies recognizing the epitope tags). NEK1: sheep SA354; third bleed; antigen corresponded to GST–NEK1 (NM_001199397.1) aa 900–1,286 expressed in bacteria. C21ORF2: sheep DA066; second bleed; antigen corresponding to maltose-binding protein–tagged C21ORF2 aa 1–256 (NM_004928.2) expressed in bacteria. Sheep were immunised with the antigens followed by four further

**Table 2.   Oligonucleotides used in this work.**

| Primer name/pair | Direction | Sequence (5′–3′) |
| --- | --- | --- |
| NEK1 exon 3 pair A (for NA and NB sgRNAs) | Fwd | ATGCAGTATTTGGCTTTCCAAAGC |
| | Rev | GTCCAAACTAGATTAGGTTACAGAACAGC |
| NEK1 exon 3 pair B (for NA and NB sgRNAs) | Fwd | TCATGTAAGTTTGCCTTTCTCTCC |
| | Rev | GATGCTTCATGTTTGCCAATACTGC |
| NEK1 exon 7 pair A (for NC sgRNAs) | Fwd | TCTGTGTGGATTGGCCTATTCTGG |
| | Rev | ACAGATTTCAGGTGACAAGTAGTATGGG |
| NEK1 exon 7 pair B (for NC sgRNAs) | Fwd | CATTCCTCTCCACCTAGACTCTGG |
| | Rev | CCTATGCAAGTTCGAGCCAGC |
| C21ORF2 exon 4 pair A (for CA and CB sgRNAs) | Fwd | TTTTCCTCACCTGTGAAGGAGGG |
| | Rev | GGGATCAAGAGGAGACACAGAAGC |
| C21ORF2 exon 4 pair B (for CA and CB sgRNAs) | Fwd | GCAACATGATATGTCCCCTGAGAGG |
| | Rev | GAGACTTCACAGGACACATCTATGTCC |

injections 28 d apart. Bleeds were performed seven days after each injection.

### Cell lines

All cell lines used in this study were derived from ARPE-19 cells. Cells were incubated at 37°C, 5% CO2, and maintained in DMEM/Nutrient Mixture F-12 (DMEM/F-12; Thermo Fisher Scientific) supplemented with 10% FBS (Thermo Fisher Scientific), 1 × 10,000 U/ml penicillin–streptomycin (Thermo Fisher Scientific), 1/100 200 nM L-glutamate (Thermo Fisher Scientific). ARPE-19 NEK1–KO and C21ORF2–KO cell lines were maintained as above with a culture medium supplemented with 20% FBS. Stably transduced ARPE-19 NEK1–KO and C21ORF2–KO cells were maintained as above with a culture medium supplemented with 20% FBS and 2 µg/ml puromycin (Thermo Fisher Scientific). All cell lines were cultured in medium supplemented with 10% FBS during the duration of experiments. U2-O-S cells DR-GFP cells and U2-O-S TLR cells were cultured in DMEM supplemented with 10% FBS (Wisent) and 1% penicillin/streptomycin (Thermo Fisher Scientific).

**Table 3.   siRNAs used in this work.**

| siRNA name | Sequence (5′–3′) |
| --- | --- |
| Luciferase | CGUACGCGGAAUACUUCGA |
| NEK1 | GGUCUGUUUGAUGCAAACAACCCAA |
| C21ORF2-2 | GCCUACAGAAGCUGGACAA |
| C21ORF2-3 | UGAGUGAGGGAGAGGAGA |
| C21ORF2-4 | GGAUGAACGUGGCCUGAAG |
| C21ORF2-5 | AGGCUGUGACGGAGGAGG |
| C21ORF2-6 | AAGCUAUGCUGCACACUGA |
| C21ORF2-7 | ACAGCGAGGAGGAGGCAAC |
| C21ORF2-8 | AGAGAGAGGGCACAGGCCA |
| C21ORF2-9 | UGCGGGAGCUGGAUGCAG |

### siRNA transfection

150,000 ARPE-19 cells were seeded in a 5-cm plate and allowed to recover for 24 h. Cells were transfected with 50 nM siRNA using Lipofectamine RNAiMAX (Thermo Fisher Scientific) according to the manufacturer's instructions. 8 h post-transfection, cells were washed once with PBS and fresh complete DMEM F/12 culture medium was added. Cells were harvested 64 h later.

### PEI transfection

For transfection of one 70% confluent ARPE-19 10 cm dish, a mixture of 10 µg cDNA, 20 µl 1 mg/ml PEI MAX (Polysciences) topped up to 1 ml with OptiMEM medium was prepared. Transfection reaction was incubated 20 min at room temperature before being added to cells in a dropwise fashion. 8 h post-transfection, cells were washed once with PBS and added fresh complete DMEM F/12 culture medium. Cells were harvested 24 h post-transfection.

### Genome editing

ARPE-19 cells were transfected with a pair of plasmids targeting exon 3 or 4 in *NEK1* sequence or exon 4 in *C21ORF2* sequence using Lipofectamine 2000 (#11668027; Thermo Fisher Scientific) according to manufacturer's instructions. 8 h post-transfection, cells were washed once with PBS and fresh complete DMEM F/12 culture medium was added. Cells were allowed to recover for additional 40 h before being selected for antibiotic resistance using complete DMEM/F12 medium supplemented with 2 µg/ml puromycin. The puromycin media was refreshed after 24 h for a total selection time of 48 h. Cells were then cultured for an additional 3–5 d to provide time for gene editing and eventually seeded at low densities (500 cells) in 15 cm dishes. Single colonies were isolated using cloning discs (Sigma-Aldrich) soaked with trypsin 2–4 wk later.

**Table 4.  Antibodies used in this work.**

| Primary antibodies | | | | | |
|---|---|---|---|---|---|
| **Antigen** | **Species** | **Manufacturer** | **Identifier** | **Dilution** | **Application** |
| FLAG | Mouse | Sigma-Aldrich | F1804 | 1:5,000 | WB |
| HA | Rabbit | CST | 3724 | 1:1,000 | WB |
| NEK1 | Rabbit | ProteinTech | 27146-1-AP | 1:1,000 | WB |
| NEK1 (aa 700–1,286) | Sheep | MRC-PPU | SA353 | 1:2,000 | WB |
| NEK1 (aa 900–1,286) | Sheep | MRC-PPU | SA354 | 1:2,000 | WB |
| NEK1 | Rabbit | Bethyl Laboratories | A304-570A | 1:2,000 | WB |
| C21ORF2 (full length) | Sheep | MRC-PPU | DA066 | 0.1 μg/ml | WB |
| C21ORF2 (full length) | Rabbit | ProteinTech | 27609-1-AP | 1:1,000 | WB |
| GAPDH | Rabbit | CST | 2118 | 1:10,000 | WB |
| α-Tubulin | Mouse | CST | 3873 | 1:10,000 | WB |
| GFP | Chicken | Abcam | Ab13970 | 1:1,000 | WB |
| HIF-1α | Mouse | BD | 610959 | 1:1,000 | WB |
| ATG14 | Rabbit | CST | 96752 | 1:1,000 | WB |
| ATG14 pSer29 | Rabbit | CST | 92340 | 1:1,000 | WB |
| LC3A/B | Rabbit | CST | 4108 | 1:1,000 | WB |
| FLAG | Mouse | Sigma-Aldrich | F1804 | 1 μg/1 mg lysate | IP |
| Mouse IgG1 (isotype control for FLAG antibody) | Mouse | CST | 5415 | 1 μg/1 mg lysate | IP |
| HA beads | Mouse | MRC-PPU | Frankenbody | 15 μl 50% slurry/ 1 mg lysate | IP |
| Mouse IgG2b (isotype control for HA beads) | Mouse | CST | 53484 | 1 μg/1 mg lysate | IP |
| NEK1 | Rabbit | ProteinTech | 27146-1-AP | 2 μg/1 mg lysate | IP |
| NEK1 (aa 900–1,286 aa) | Sheep third bleed | MRC-PPU | SA354 | 1 μg/1 ml lysate | IP |
| NEK1 | Rabbit | Bethyl Laboratories | A304-570A | 2 μg/1 mg lysate | IP |
| C21ORF2 (full length) | Sheep fourth bleed | MRC PPU | DA066 | 1 μg/1 mg lysate | IP |
| C21ORF2 (full length) | Rabbit | ProteinTech | 27609-1-AP | 2 μg/1 mg lysate | IP |
| Rabbit IgG | Rabbit | CST | 2729 | 2 μg/1 mg lysate | IP |
| Sheep IgG | Sheep | Abcam | ab37385 | 1 μg/mg lysate | IP |
| NEK1 (aa 900–1,286) | Sheep | MRC PPU | SA344 | 1:100 | IF |
| C21ORF2 (full length) | Sheep | MRC PPU | DA066 | 1:200 | IF |
| GTU88 (γ-tubulin) | Mouse | Sigma-Aldrich | T6557 | 1:500 | IF |
| ɣH2AX | Mouse | Merck Millipore | 05-636 | 1:2,000 | IF |
| ARL13B | Rabbit | ProteinTech | 17711-1-AP | 1:200 | IF |
| Pericentrin | Mouse | Abcam | ab28144 | 1:500 | IF |
| Secondary antibodies | | | | | |
| Antigen | Species | Manufacturer | Code | Dilution | Application |
| Anti-mouse 680 | Donkey | LI-COR | 926-68072 | 1:10,000 | WB |
| Anti-mouse 800 | Donkey | LI-COR | 926-32212 | 1:10,000 | WB |
| Anti-rabbit 680 | Donkey | LI-COR | 926-68073 | 1:10,000 | WB |
| Anti-rabbit 800 | Donkey | LI-COR | 926-32213 | 1:10,000 | WB |
| Anti-sheep AF680 | Donkey | Thermo Fisher Scientific | A21102 | 1:10,000 | WB |
| Anti-sheep Dylight 800 | Donkey | Rockland | 613-745-168 | 1:10,000 | WB |

| Primary antibodies | | | | | |
|---|---|---|---|---|---|
| **Antigen** | **Species** | **Manufacturer** | **Identifier** | **Dilution** | **Application** |
| Anti-chicken 800 | Donkey | LI-COR | 926-32218 | 1:10,000 | WB |
| Protein G Dylight 800 | n/a | Rockland | PG00-45 | 1:10,000 | WB |
| Anti-mouse AF488 | Donkey | Thermo Fisher Scientific | A21202 | 1:1,000 | IF |
| Anti-mouse AF647 | Donkey | Thermo Fisher Scientific | A32787 | 1:1,000 | IF |
| Anti-mouse AF647 | Goat | Thermo Fisher Scientific | #A21235 | 1:500 | IF |
| Anti-rabbit AF488 | Donkey | Thermo Fisher Scientific | A21206 | 1:1,000 | IF |
| Anti-rabbit AF647 | Donkey | Thermo Fisher Scientific | A31573 | 1:1,000 | IF |
| Anti-sheep AF488 | Donkey | Thermo Fisher Scientific | A11015 | 1:500 | IF |

**Retrovirus production for stable expression of target proteins**

To reintroduce expression of WT or mutated versions of NEK1/C21ORF2, NEK1, or C21ORF2–KO cells were infected with lentiviruses. HEK293FT cells were transfected with plasmids encoding for proteins of interest along with the GAG/Pol and VSVG constructs required for lentiviral production. PEI was used as a transfection reagent. 48 h later, lentiviral-containing medium was collected, filtered through a 0.22-$\mu$m pore filter (Millipore), and supplemented with polybrene (8 $\mu$g/ml). Cells were transduced for 24 h followed by a 48-h selection with 2 $\mu$g/ml puromycin. Successful lentiviral integration was confirmed through Western blotting.

**Cell lysis and immunoprecipitation**

Cells were lysed in ice cold buffer comprising 50 mM Tris–HCl (pH = 7.4), 150 mM NaCl, 270 mM sucrose, 1 mM EGTA (pH 8.0, Sigma-Aldrich), 10 mM $\beta$-glycerol phosphate disodium salt pentahydrate (Sigma-Aldrich), 5 mM sodium pyrophosphate decahydrate (Sigma-Aldrich), 50 mM sodium fluoride (Sigma-Aldrich), 10 ng/ml microcystin-LR (Sigma-Aldrich), 0.5 U/ml Pierce Universal Nuclease (Thermo Fisher Scientific), 1:100 phosphatase inhibitor cocktail 2 (Sigma-Aldrich), 1x Complete EDTA-free protease inhibitor (Roche), 1 mM sodium orthovanadate (Sigma-Aldrich), 1 mM AEBSF, 1 mM benzamidine, 10 mM iodoacetamide (Sigma-Aldrich), and 1% (vol/vol) Triton X-100 (Sigma-Aldrich). Lysates were incubated on ice for 20 min followed by centrifugation at 14,000$g$, 4°C for 15 min. Proteins were quantified using the BCA Kit (Thermo Fisher Scientific). First, lysates were pre-cleared with protein G Sepharose beads conjugated to a primary antibody isotype control for 1 h at 4°C with shaking. Lysates were then transferred to tubes containing protein G Sepharose beads conjugated to primary antibodies. Reactions were incubated for 1.5 h at 4°C with shaking. Protein-bound beads were washed 5 × 15 min at 4°C on a rotating wheel with a buffer composed of 50 mM Tris (pH = 7.4), 500 mM NaCl, and 1% Triton X-100

before being analysed by SDS–PAGE and Western blotting. The in-house MRC-PPU–generated NEK1 and C21ORF2 were used in all relevant immunoprecipitation experiments.

**SDS–PAGE and Western blotting**

The proteins were separated by SDS–PAGE using 4–12% Bis/Tris gels (Thermo Fisher Scientific) under reducing conditions for 3 h at 90 V. NuPAGE 3-[N-morpholino] propane sulphonic acid (MOPS) running buffer was used (Thermo Fisher Scientific). Proteins were transferred to 0.45 $\mu$M nitrocellulose membrane (Cytiva) at a constant voltage of 90 V for 1.5 h. Membranes were blocked for 1 h at room temperature using 5% milk/TBST. Detection of antigens was carried out using antibodies specified in Table 4. All antibodies were dissolved in 5% milk/TBST and incubated overnight (at least 16 h) at 4°C. Subsequently, membranes were washed 3 × 5 min using TBST. Primary antibodies were detected using secondary antibodies specified in Table 4. All secondary antibodies were dissolved in 5% milk/TBST and incubated for 1 h at room temperature. Membranes were washed 3 × 10 min using TBST and 1 × 10 min using PBS. Data were acquired using an Odyssey CLx LI-COR scanner and analysed in Image Studio v. 5.2. The in-house MRC-PPU–generated NEK1 and C21ORF2 were used in all Western blotting experiments except for blotting gel filtration fractions in Fig 1C where Bethyl Laboratories NEK1 antibodies were used (see Table 4 above).

**Gel filtration of whole lysates**

Ten 70% confluent 10 cm culture dishes containing ARPE-19 cell lines were lysed and 500 $\mu$l total cell extract was loaded onto the Superose 6 Increase 10/300 column which was equilibrated in gel filtration buffer. 0.5 ml fractions collected were analysed by SDS–PAGE and Western blotting. Gel filtration standard from Bio-Rad was used (#151–1,901).

## Immunodepletion following gel filtration

For one reaction, 50 µl 50% slurry Sepharose G was conjugated with 5 µg primary anti-NEK1 (900–1,286, MRC-PPU), 5 µg anti-C21ORF2 (full length, MRC-PPU), or 5 µg sheep isotype control (Abcam) antibodies. Indicated samples following gel filtration of whole ARPE-19 lysates section were pooled and divided into three equal parts. One part was used for NEK1 immunodepletion, one part for C21ORF2 immunodepletion, and one part was added to sheep isotype control-bound beads; this served as a negative control. Each part of lysate was subjected to three rounds of immunodepletion. Each immunodepletion reaction was incubated for 1.5 h at 4°C with applied shaking. Protein-bound beads were washed 3 times with 50 mM Tris pH = 7.4 buffer supplemented with 500 mM NaCl and 1% Triton X-100 before being analysed by SDS–PAGE and Western blotting.

## Immunofluorescence experiments

### Ciliogenesis studies
60,000 cells were seeded in an eight-chamber slide (Ibidi) and allowed to recover for 24 h. Subsequently, cells were washed extensively with PBS to remove any residual traces of FBS. Ciliogenesis was induced by culturing cells with OptiMEM for up to 48 h. Cells were stained with relevant antibodies using a protocol as in the antibody staining section. Data were acquired using a Leica SP8 confocal microscope equipped with a white laser. A 63x, 1.2 numerical aperture objective was used, and data were analysed in Fiji ImageJ v.1.53 and Omero v.5.5.17. A cell has been counted a ciliated if a clear co-localisation of pericentrin and ARL13B was detected. Mitotic cells were excluded from the analysis. Cells were washed with PBS followed by fixation with 3% PFA/PBS for 15 min at room temperature. Subsequently, samples were washed twice with PBS before permeabilisation with 0.2% Triton X-100/PBS for 5 min at room temperature. For NEK1 and C21ORF2 localisation studies, cell fixation was followed by a 5 min incubation at −20°C with pre-cooled methanol (Sigma-Aldrich). If detecting EdU, samples were added Click-iT buffer (PBS supplemented with 10 mM L-ascorbic acid, 2 mM CuSO$_4$, and Alexa Fluor 647 azide [Thermo Fisher Scientific] at 1,875 µM) for 60 min at room temperature in the dark. Cells were washed twice with PBS before being blocked with DMEM (Thermo Fisher Scientific) supplemented with 10% FBS for 30 min at room temperature in the dark. Cells were incubated with primary antibodies diluted in the blocking buffer for 90 min at room temperature in the dark. Detection of antigens was carried out using antibodies specified in Table 4. Cells were washed three times with PBS before being added secondary antibodies diluted in blocking buffer for 60 min at room temperature in the dark. Detection of primary antibodies was carried out using secondary antibodies specified in Table 4. Subsequently, cells were washed three times with PBS. Nuclei were visualised by staining with 1 mg/ml DAPI for 15 min at room temperature in the dark. Consequently, cells were washed once with PBS and stored at 4°C.

### NEK1 and C21ORF2 localisation studies
For testing the localization of NEK1 and C21ORF2 in cycling cells (Fig 5A), cells were seeded on coverslips (Standard #1.5; Thermo

Fisher Scientific) in a 24-well plate at a density of 4 × 10$^4$ cells/well for ARPE-19 WT or 6 × 10$^4$ cells/well for ARPE-19 NEK1 or C21ORF2–KO cell lines. Cells were grown for 24 h before fixation. For cilia localization, ARPE-19 WT cells (6 × 10$^4$ cells/well) were seeded to an eight-well Nunc Lab-Tek Chambered Coverglass (#155361; Thermo Fisher Scientific) for 24 h and serum starved with FBS-free culture medium for 48 h before fixation. For fixation, cells were washed with PBS, fixed for 3 min at room temperature in 3% PFA (Acros Organics) in PBS followed by 5 min permeabilisation at −20°C with pre-cooled methanol (Sigma-Aldrich). Cells were washed three times with PBS before staining. Fixed cells were blocked with 3% BSA (A9647; Sigma-Aldrich) in 0.1% (vol/vol) Triton X-100 (Sigma-Aldrich)/PBS (hereafter called PBX) for 30 min. Samples were incubated with primary antibodies diluted in 3% BSA in PBX for 1 h at room temperature in a humid chamber. After washing three times with PBX, samples were incubated with secondary antibodies in a humid chamber for 30 min. Cells were washed three times with PBS and mounted with Mowiol (Calbiochem).

## Mass spectrometry immunoprecipitation experiments

### Immunoprecipitation
Sepharose beads were washed 3x with PBS before conjugation with antibodies. For one reaction, 50 µl 50% slurry Sepharose G was conjugated with 5 µg primary anti-NEK1 (900–1,286, MRC-PPU) or anti-C21ORF2 (full length, MRC-PPU) antibodies. Cells were lysed using mammalian lysis buffer, and 5 mg of lysate was used per each IP. Reactions were incubated for 1.5 h at 4°C with applied shaking. Protein-bound beads were washed three times with PBS supplemented with 0.5% NP-40. Protein-bound beads were stored in −20°C.

### Protein digestion
Proteins were eluted from the beads with 23 µl elution buffer (5% SDS, 50 mM TEAB in water), diluted with 165 µl binding buffer (100 mM TEAB [final] in 90% LC-grade methanol), and loaded on an S-Trap micro cartridge. Proteins were concentrated in the S-Trap cartridge by centrifugation at 4,000 g for 1 min and washed five times with 150 µl binding buffer. Proteins were then digested for 2 h at 47°C by adding 20 µl 100 mM TEAB supplemented with 5 µg trypsin (Thermo Fisher Scientific). Digested peptides were then eluted from the S-Trap by sequential addition of 40 µl 50 mM TEAB, 40 µl 0.2% formic acid in LC-grade water, and 40 µl 50% acetonitrile/50% LC-grade water. Eluted peptides were dried down by SpeedVac.

### TMT labelling
Dried peptides were reconstituted in 50 µl of 100 mM TEAB and labelled by adding 10 µl of TMT reagent (Thermo Fisher Scientific) at 19.5 µg/µl. After 2 h under agitation at room temperature, the reaction was quenched by adding 5 µl of 5% hydroxylamine for 15 min. Labelled peptides were then dried down by SpeedVac.

### Offline fractionation
Peptide were resuspended in 100 µl of 5 mM ammonium acetate at pH 10 and injected on an XBridge Peptide BEH C18 column (1 mm ×

100 mm, 3.5 µm particle size, 130 Å pores, #186003561; Waters). Peptides were eluted from the column using basic reverse phase fractionation (using 10 mM ammonium acetate in LC-grade water as buffer A and 10 mM ammonium acetate in 80% acetonitrile/20% LC-grade water as buffer B) on a 55 min multistep gradient at 100 µl/min (from 3 to 10%, 40%, 60%, and 100% buffer after at 20, 25, 65, 70, and 75 min, respectively). Eluted peptides were collected from 26 to 82 min into 96 fractions and pulled into 24 fractions using non-consecutive concatenation (fraction 1 was pulled with 25, 49, and 73). The 24 fractions were then dried down by SpeedVac and stored at –20°C until LC–MS/MS analysis.

### LC–MS/MS analysis

Fractionated peptides were resuspended in 20 µl 5% formic acid in water and injected on an UltiMate 3000 RSLCnano system coupled to an Orbitrap Fusion Lumos Tribrid Mass Spectrometer (Thermo Fisher Scientific). Peptides were loaded on an Acclaim Pepmap 100 trap column (100 µm × 2 cm, 5 µm particle size, 100 Å pores, #164564-CMD; Thermo Fisher Scientific) for 5 min at 10 µl/min prior analysis on a PepMap RSLC C18 analytical column (75 µm × 50 cm, 2 µm particle size, 100 Å pores, #ES903; Thermo Fisher Scientific) and eluted on a 115 min linear gradient from 3 to 35% buffer B (buffer A: 0.1% formic acid in LC-grade water, buffer B: 0.08% formic acid in 80% acetonitrile/20% LC-grade water). Eluted peptides were then analysed by the mass spectrometer operating in synchronous precursor selection mode on a TOP 3 s method. MS1 were recorded at a resolution of 120,000 at m/z 200 using an automatic gain control (AGC) target of 100% and a maximum injection time (IT) of 50 ms. Precursors were selected in a data-dependant manner using an AGC target of 100% and maximum IT of 50 ms for MS2 fragmentation using HCD at a normalised collision energy of 35% and analysed in the ion trap operating in rapid mode. For synchronous precursor selection mode, up to 10 fragment ions were selected for MS3 fragmentation using an AGC target of 200%, a maximum IT of 100 ms and a normalised collision energy of 65%. MS3 fragments were then analysed in the Orbitrap using a resolution of 50,000 at m/z 200.

### Data analysis

Peptide search against the UniProt–SwissProt Human database (released on 05/10/2021) using MaxQuant 1.6.17.0 in MS3 reporter ion mode using default parameters with the addition of Deamidation (NQ) and Phospho (STY) as variable modification. Statistical analysis was carried out using Python (v3.9.0) and the packages Pandas (v1.3.3), Numpy (v1.19.0) and SciPy (v1.7.1). In short, protein groups only identified by site, from the reversed or potential contaminants database, identified with less than 2 razor or unique peptides and quantified in less than four out of five replicates were excluded. Missing values were then imputed using a Gaussian distribution centred on the median with a downshift of 1.8 and width of 0.3 (based on the SD) and protein intensities were median normalised. Protein regulation was assessed using a two sample Welch test and P-values were adjusted using Benjamini Hochberg (BH) multiple hypothesis correction. Proteins were considered significantly regulated if the BH-corrected

P-value was smaller than 0.05 and the old change was greater than 2 or smaller than 0.5.

The mass spectrometry data relating to Fig 2 have been deposited to the ProteomeXchange consortium via the PRIDE (Perez-Riverol et al, 2022) partner repository with the dataset identifier PXD036410. The raw data for Fig 2A, are also given in Table S1, and the raw data for Fig 2B, are given in Table S2.

### AlphaFold modelling

Modelling of NEK1 (aa1,160–1,286, UniProt: Q96PY6-3) and C21ORF2 (aa1-256, UniProt: O43822-1) complex binding was performed using AlphaFold docking (Jumper et al, 2021), using ColabFold (Mirdita et al, 2022) and the AlphaFold2.ipynb notebook with default settings, using MMseqs2 for sequence alignment and AlphaFold2-multimer-v2 model with AMBER structure relaxation to ensure appropriate orientation of the side chains to avoid steric clashes. The predicted structures of the five models obtained were visualised using PyMOL and aligned with the Alignment/Superposition plugin. Intermolecular interactions for each model were predicted using BIOVIA Discovery Studio Visualizer 2021 with the Non-bond Interaction Monitor and are provided in the supplementary file (Supplemental Data 1).

### DR-GFP

The DR-GFP assay (Pierce et al, 1999) was carried out in a U2-O-S derivative containing a copy of the DR-GFP reporter (U2-O-S DR-GFP; Nakanishi et al, 2011) in which cells were transfected with 10 nM siRNA (Dharmacon) using Lipofectamine RNAiMAX (Thermo Fisher Scientific). 48 h post-transfection, cells were transfected a second time with 2 µg pCBASceI plasmid (#26477; Addgene) using PEI (Polysciences). 48 h post-transfection, cells were trypsinised, and the percentage of GFP-expressing cells was assessed using an Attune NxT flow cytometer (Thermo Fisher Scientific).

### TLR assay

The TLR assay (Certo et al, 2011) was carried as follows. U2-O-S cells were transduced with lentiviral particles prepared with pCVL.TrafficLightReporter.Ef1a.Puro (#31482; Addgene) at a low multiplicity of infection and selected with 2 µg/ml puromycin (Gibco). The resulting U2-O-S TLR cells were transfected with 10 nM siRNA (Dharmacon) using Lipofectamine RNAiMAX (Thermo Fisher Scientific). 48 h post-transfection, $10^6$ cells were nucleofected with 5 µg of pCVL.SFFV.d14mClover.Ef1a.HA.NLS.Sce(opt).T2A.TagBFP (#32627; Addgene) in 100 µl electroporation buffer (Amaxa Cell Line Nucleofector Kit V) using program X001 on a Nucleofector II (Amaxa). 72 h post-nucleofection, GFP and mCherry fluorescence were measured in BFP-positive cells using a Fortessa X-20 flow cytometer (BD Biosciences).

### Cell cycle analysis (U2-O-S DR-GFP cells)

The cell cycle analysis was performed in U2-O-S DR-GFP (Nakanishi et al, 2011). Cells were transfected with 10 nM siRNA (Dharmacon) using Lipofectamine RNAiMAX (Invitrogen). 72 h post-transfection, cells were pulsed with 20 µM EdU (5-ethynyl-2′-deoxyuridine; Thermo Fisher Scientific) for 30 min. Cells were then trypsinised, washed, and then fixed in 4% paraformaldehyde (Thermo Fisher Scientific). After fixation, samples were washed in PBS-B (1% bovine serum albumin fraction V in PBS filtered through 0.2 µm membrane) and then permeabilised at room temperature for 15 min by resuspending the pellets in PBS-B/ 0.5% Triton X-100 (Sigma-Aldrich). Cell pellets were rinsed with PBS-B and incubated with EdU staining buffer containing 150 mM Tris–Cl, pH 8.8, 0.1 mM CuSO4, 100 mM ascorbic acid, and 10 µM Alexa Fluor 555 azide (Thermo Fisher Scientific) in water for 30 min at room temperature. After rinsing with PBS-B, cells were resuspended in the analysis buffer (PBS-B containing 0.5 µg/ml DAPI and 250 µg RNase A) and incubated 30 min at 37°C. Cells were stained with DAPI (0.8 µg/ml) and analysed using an Attune NxT flow cytometer (Thermo Fisher Scientific), recording at least 20,000 events and analysed using FlowJo v10.

### Schematic diagrams

Schematic diagrams of C21ORF2 and NEK1 were prepared using IBS software (Liu et al, 2015).

### Statistics and data reproducibility

No statistical method was used to predetermine sample size, and investigators were not blinded to allocation during the performance of experiments and assessment of results.

Graphs were generated and statistical tests performed using Prism 9 software (https://www.graphpad.com/scientific-software/prism/; GraphPad), as described above and in the figure legends. A one-way ANOVA test with multiple comparisons was used to analyse the data in Fig 5E.

## Data Availability

The mass spectrometry data relating to Fig 2 will have been deposited to the ProteomeXchange consortium via the PRIDE (Perez-Riverol et al, 2022) partner repository with the dataset identifier PXD036410. AlphaFold output files have been uploaded to Zenodo and can be accessed at the following link: Pawel, 2022.

## Supplementary Information

## Acknowledgements

We thank the technical support of the MRC-PPU including the DNA Sequencing Service, Tissue Culture team, Reagents and Services team, and the PPU Mass Spectrometry team. We thank Axel Knebel for help with gel filtration and protein purification. We are grateful to Luis Sanchez-Pulido and Chris Ponting for help with bioinformatic analyses. We are grateful to Frank Zenke, Ulrich Pehl, and Claudio Lademann from Merck KGaA, Darmstadt for useful discussions. We thank Dominika Pakiela for help with drawing the model in Fig 6C. We thank Constance Alabert for help with microscopy. We are grateful to Matthew McFarland for help with analysis of primary cilia. We thank Florian Weiland and members of the Rouse team for useful discussions. We thank Rathan Jadav for help with microscopy. This work was supported by the Medical Research Council (grant number MC_UU_12016/1; M Gregorczyk, P Lis, F Lamoliatte, I Muñoz, J Rouse), by a grant from the Canadian Institutes for Health Research (grant PJT 180438 to D Durocher) and by Merck KGaA (Florian Weiland and J Rouse).

### Author Contributions

M Gregorczyk: formal analysis, investigation, and visualization.
G Pastore: investigation and methodology.
I Muñoz, T Carroll, J Streubel, M Munro, G Pereira, and T Macartney: investigation.
P Lis, S Lange, and F Lamoliatte: formal analysis and investigation.
R Toth: resources.
F Brown: conceptualization, resources, and methodology.
J Hastie: resources and methodology.
D Durocher: formal analysis and investigation.
J Rouse: conceptualization, formal analysis, supervision, funding acquisition, and methodology.

### Conflict of Interest Statement

D Durocher is a shareholder and advisor of Repare Therapeutics.

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
