## [Reviewer comments · Life Science Alliance]

Life Science Alliance

Functional characterization of C21ORF2 association with the NEK1 kinase mutated in human in disease

Mateusz Gregorczyk, Graziana Pastore, Ivan Muñoz, Thomas Carroll, Johanna Streubel, Meagan Munro, Pawel Lis, Sven Manfred Lange, Frederic Lamoliatte, Thomas Macartney, Rachel Toth, Fiona Brown, C. James Hastie, Gislene Pereira, Daniel Durocher, and John Rouse

DOI: <https://doi.org/10.26508/lsa.202201740>

Corresponding author(s): John Rouse, University of Dundee and John Rouse, University of Dundee

Review Timeline:

Submission Date:	2022-09-27
Editorial Decision:	2022-10-24
Revision Received:	2023-03-13
Editorial Decision:	2023-04-11
Revision Received:	2023-04-24
Accepted:	2023-04-25

Transaction Report:

October 24, 2022

Re: Life Science Alliance manuscript #LSA-2022-01740

Prof. John W Rouse
University of Dundee
MRC Protein Phosphorylation and Ubiquitylation Unit
Hawkhill
Dundee DD1 5EH
United Kingdom

Dear Dr. Rouse,

Thank you for submitting your manuscript entitled "Interaction with C21ORF2 controls the cellular functioning of the NEK1 kinase" to Life Science Alliance. The manuscript was assessed by expert reviewers, whose comments are appended to this letter. We invite you to submit a revised manuscript addressing the Reviewer comments.

Thank you for this interesting contribution to Life Science Alliance. We are looking forward to receiving your revised manuscript.

Sincerely,

B. MANUSCRIPT ORGANIZATION AND FORMATTING:

Reviewer #1 (Comments to the Authors (Required)):

The paper by Gregorczyk et al. explores to understand better the function of the NEK1 kinase in human cells. NEK1 is a kinase that, in humans, is mutated in SMD and ALS. It is, though not very clear what is the specific function of NEK1 and what are the specific interactors. The authors explored expanding our understanding of NEK1 by performing molecular experiments to understand what are the NEK1 interactors and which residues are important for such interactions. They show that NEK1 is in a complex with C21ORF2 (itself mutated in SMD and ALS) via the CID domain. To explore what is the functional importance of this complex they mutate both NEK1 and C21ORF2 interaction residues and show that, outside of the known mitosis function, cells that cannot form this complex have defects in HR and ciliogenesis.

The paper brings important new information on NEK1 and merits its publication upon minor corrections. I note that important resources (antibodies) were generated and nicely validated.

Minor suggestions:

1. One of the first improvements would be on the writing of the paper:

- many parts of the results have paragraphs that could altogether be moved to methods or the figure legends (e.g. on page 6, statements on how many clones were picked; on page 7, "Five biological replicates were used...."; on page 11 "Transfection of TLR..."; here figure 5B can be moved to suppl.)

- in some parts, there is unneeded repetition (e.g. in the Discussion section on page 14, is stated, "It is interesting to note that the symptoms associated with SMD, caused by NEK1 or C21ORF2 mutations, for example, are somewhat reminiscent of the SPONASTRIME syndrome caused by mutations in TONSL that weaken HR (Burrage et al., 2019; Chang et al., 2019). It is possible, therefore, that SMD symptoms are caused by defective HR, and this idea will be important to test." And then, on page 15, in the final paragraph, "The SMD symptoms caused by NEK1 mutations are reminiscent of those reported in SPONASTRIME syndrome caused by mutations in the TONSL HR gene (Burrage et al., 2019; Chang et al., 2019). Therefore, it is possible that it is the HR function of NEK1 that is responsible for preventing SMD."

- few typos (e.g. page 7, last paragraph, "Nek1-KO cells were to control for non-specific binding"; probably a verb is needed ("were used to control...")

- the section on structure-function-based phenotypic analysis would fit better after the AlphaFold

2. Throughout, there are a lot of statements of data unshown, and some of this data would be critical to show (for example, GFP-C21ORF2 PD and FLAG-NEK1 blotting; RAD54 phosphorylation data etc.); I believe now most journals would not allow "data not shown" statements as all data is important.

3. On the RAD51 foci, there is quantification, but there are no representative images of the foci that would be important to be shown.

4. Figure 2 shows the mass-spec data, but I couldn't find the associated files; even though the statement on the availability of raw data is given, the authors should make the stats files available as Data files.

5. The defect on HR with no RAD51/RAD54 data is intriguing. The authors should try to propose an explanation in the Discussion. The data showing the HR defect is done in U2OS, while the foci and S-527 RAD54 are in ARPE19. While the authors show in Fig. S5 that NEK1/C21ORF2 interaction happens in other human cell lines, this experiment doesn't prove that there is the same dependency on each member for the interaction. Would be important to know if upon siNEK1 or siC21ORF2 in U2OS, the complex also disassembles; Watanabe et al. 2020 iScience show, for example, that in HEKS KO of C21ORF2 reduces but does not completely destabilise NEK1 levels. Alternatively, look at S-527 RAD54 or RAD51 foci in U2OS upon NEK1 siRNA depletion. Maybe a limitation of the study section could be added to address some of the limitations (another example is the use of only 1 NEK1 KO clone).

6. extreme defect on cilia formation; from the images, it seems that pericentrin is not affected, but the ARL13B is disassembled (green dispersed dots can be observed in NEK1 KO but not parental). Is NEK1 interacting with ARL13B or affecting its Y406

phosphorylation; I note that NEK1 is a mainly T-kinase but in certain conformations (see Bert van de Kooij et al. eLife. 2019; doi: 10.7554/eLife.44635; a paper that I feel should be cited in the introduction at least) its kinase activity can shift to Y ("APE-4 isoleucine in Nek1 enhances substrate phosphorylation on tyrosine residues"; the mutation is NEK1 P168I).

Reviewer #2 (Comments to the Authors (Required)):

The informative manuscript by Gregorczyk et al. described that C21ORF2 and NEK1 kinase interact each other and their binding regulates generation of cilia. Both of these proteins were reported to associate with ALS or ciliopathies, suggesting that these proteins function on the same pathway.

Authors generated KO cells of NEK1 as well as C21ORF2 and observed phenotypes of these cells and tried to rescue them with wild type or mutant cDNA to identify functional domain of these proteins.

These findings were quite informative for functional comprehension of NEK1 and C21ORF2, but there seems to be long distance between biochemical function and pathological phenotype. This reviewer could not logically trace biological significance of binding of NEK1 and C21ORF2 and have ideas to solve pathological symptoms.

This reviewer suggests that it is necessary to add some data as well as careful discussion to be published in Life Science Alliance.

Major points

1) Authors decided binding domains of NEK1 and C21ORF2, predicted more detail using AlphaFold, and validated using mutated protein. Unfortunately, it is hard to say that this information opens a new incite about biological or pathological function or NEK1 and C21ORF2.

2) Authors described suppression of NEK1 expression decreased C21ORF2 expression. They tried some reagents for regulation of protein stability, but failed to elucidate that mechanism. To clarify the function of NEK1 or C21ORF2, authors should find suggestive mechanisms, also, should show the effect of this decreased expression on phenotype of cells. It is very understandable if there are data showing whether the effect of NEK1 KO on HR and ciliogenesis is rescued or not by C21ORF2 overexpression.

3) In this article, authors described the functions of NEK1 and C21ORF2 were observed during HR and ciliogenesis. If so, where do NEK1 and C21ORF2 exist?

4) Author used cells derived from ARPE-19 cells. This cell line is derived from retinal pigment epithelia (RPE) cell, and frequently is used to observe ciliogenesis. But as authors described some pathogenesis of NEK1 and C21ORF2 is related to neuron. Whether what authors observed here also observed in neuron?

Minor points

5) In abstract, authors do not describe HR as well as ciliogenesis, but this review supposed relationship between binding affinity and phenotype is quite important points of this articles.

6) In section of introduction author focus on topics related to these results. Author just described review of these area.

Reviewer #3 (Comments to the Authors (Required)):

In this manuscript, the authors describe the generation and characterization of polyclonal antibodies and knock-out ARPE-19 cells for NEK1 and C21ORF2. These two proteins are implicated in an overlapping range of human diseases and have previously been reported to interact in mammalian cells (e.g. Wheway et al. 2015, Watanabe et al. 2020). The purpose of this current study is to use the novel reagents they have generated to extend our understanding of the nature and potential function of this interaction.

Evidence is provided here that the endogenous NEK1 and C21ORF2 form a complex in ARPE-19 cells, although in contrast to the previous report that described a third component of the complex, SPATA7, the authors argue that, at least in ARPE-19 cells, this complex is mostly devoid of other proteins. They then use AlphaFold-based modelling and biochemical experiments to map the domains of interaction and show that pathological mutations in these regions disrupt interaction. This is interesting and provides the basis for further biophysical and structural studies. Loss of interaction with NEK1 for the C21ORF2 mutations has previously been reported, but description of loss of interaction with C21ORF2 of the NEK1 mutations is novel, and this is the first time these results have been put in the context of a structural model for their mode of interaction.

The authors also attempt to explore the function of the NEK1:C21ORF2 interaction. However, while similar consequences are shown upon siRNA-mediated depletion of either NEK1 or C21ORF2 on homologous recombination in U2OS cells, and on ciliogenesis in the ARPE-19 knock-out cells, this does not prove that it is assembly of the NEK1:C21ORF2 complex that is required for these functions. The final piece of data showing that ciliogenesis is not rescued by add-back of a NEK1 mutant

(D1277A) or C21ORF2 mutant (L224P) that are defective in binding the respective partner does begin to address the importance of complex formation. However, more extensive work that is beyond the scope of the current manuscript is required to rule out the possibility that these mutations act through other mechanisms and to test in detail how these two proteins regulate each other.

Overall, I found the manuscript well written and the data to be of very high quality. However, I feel that they need to tone down some of their conclusions about the function of the NEK1:C21ORF2 complex and provide more clarity throughout the manuscript about the previously published work, explaining where their results confirm, complement or differ from previous findings. I also feel the title is not appropriate as it implies that interaction with C21ORF2 controls the cellular functions of NEK1, when the current manuscript falls some way short of showing this.

There are a series of other corrections that are required as listed below.

- The Introduction, on page 4, states that "Most ALS-associated NEK1 mutations are heterozygous and are presumed to be dominant and loss-of-function in nature". This needs some explanation as most loss-of-function mutations act in a recessive manner.
- On page 7, the text relating to Fig. 1B should indicate the percentage of Triton X-100 used in the wash IP buffers, alongside the concentration of NaCl.
- On page 7, the text describing Fig. 1C should note and explain why commercial NEK1 antibodies were used for blotting the gel filtration samples as opposed to their in-house NEK1 antibodies.
- The text talks in a couple of places about "resting, asynchronous cells"; it's not clear what it meant by "resting" and I suggest this is revised.
- On page 8, the text describing Fig 3C states "However, the S1036* and D1277A NEK1 mutants showed a dramatic reduction in their interaction with C21ORF2, and the reverse co-immunoprecipitation experiment confirmed these findings (Fig. 3C)." I couldn't find the reverse Co-IP described here?
- On page 9, there is a typo at the end of the section: L224P (text says L22P).
- It's not clear why the text discussing the AlphaFold model of the NEK1-CID indicates this to be residues 1216-1282 while Fig. 4A shows residues 1208-1286.
- On page 10 and in Fig. 4D, the C21ORF2 combined mutant is sometimes referred to as the 3KE mutant and sometimes as the 3KRE mutant. I suggest it be consistently called the 3KRE mutant.
- On page 11, there is reference to Fig. 5C in one place when I think it should be 5E?
- On page 12, it states "However, the C21ORF2 interaction defective NEK1 mutants did not rescue NEK1 expression" when I think it should say "did not rescue C21ORF2 expression".
- The last section on page 12 should also describe the results with the C21ORF2-L224P mutant.
- The Discussion needs revision taking account of the points made above, including drawing more circumspect conclusions about the function of the NEK1:C21ORF2 complex.
- The electrophoretic mobility shift detected for C21ORF2 when co-expressed with NEK1 in Fig. 3C is briefly mentioned in the Discussion, but should also be mentioned in the Results as this is an important observation.
- On Figure 5C and E, it appears that the assay used is incorrectly labelled as I believe Fig. 5C is the TLR assay and Fig. 5E is the DR-GFP assay.
- Fig. S1 and S6A differ in what is labelled as the 'acidic region' of NEK1. This should be consistent. The legend to Fig. S1 should also explain the orange and purple boxes in NEK1, and the blue and green boxes in C21ORF2.

Responses to Reviewers' Comment

We sincerely thank the Reviewers for their comments which we found insightful. We have addressed the comments in detail, modifying the text and providing extra data. The resulting changes have improved the manuscript significantly.

Reviewer 1

Minor suggestions:

1. One of the first improvements would be on the writing of the paper:

- many parts of the results have paragraphs that could altogether be moved to methods or the figure legends (e.g. on page 6, statements on how many clones were picked; on page 7, "Five biological replicates were used..."; on page 11 "Transfection of TLR..."; here figure 5B can be moved to suppl.)

We have modified the paper accordingly.

- in some parts, there is unneeded repetition (e.g. in the Discussion section on page 14, is stated, "It is interesting to note that the symptoms associated with SMD, caused by NEK1 or C21ORF2 mutations, for example, are somewhat reminiscent of the SPONASTRIME syndrome caused by mutations in TONSL that weaken HR (Burrage et al., 2019; Chang et al., 2019). It is possible, therefore, that SMD symptoms are caused by defective HR, and this idea will be important to test." And then, on page 15, in the final paragraph, "The SMD symptoms caused by NEK1 mutations are reminiscent of those reported in SPONASTRIME syndrome caused by mutations in the TONSL HR gene (Burrage et al., 2019; Chang et al., 2019). Therefore, it is possible that it is the HR function of NEK1 that is responsible for preventing SMD."

We have modified the paper accordingly.

- few typos (e.g. page 7, last paragraph, "Nek1-KO cells were to control for non-specific binding"; probably a verb is needed ("were used to control...")?)

We have rectified these typos.

- the section on structure-function-based phenotypic analysis would fit better after the AlphaFold

We tried this arrangement by swapping Figures 5 and 6 and it worked well.

2. Throughout, there are a lot of statements of data unshown, and some of this data would be critical to show (for example, GFP-C21ORF2 PD and FLAG-NEK1 blotting; RAD54 phosphorylation data etc.); I believe now most journals would not allow "data not shown" statements as all data is important.

This is a fair point. We have decided to omit reference to our RAD54 experiments, and we have removed all "data not shown" statements in the paper.

3. On the RAD51 foci, there is quantification, but there are no representative images of the foci that would be important to be shown.

We propose to remove the RAD51 data from the paper, please see response to point 5 below.

4. Figure 2 shows the mass-spec data, but I couldn't find the associated files; even though the statement on the availability of raw data is given, the authors should make the stats files available as Data files.

We have now included supplementary tables as requested: Table S1 for Fig. 2A, and Table S2 for Fig. 2B.

5. The defect on HR with no RAD51/RAD54 data is intriguing. The authors should try to propose an explanation in the Discussion. The data showing the HR defect is done in U2OS, while the foci and S-527 RAD54 are in ARPE19. While the authors show in Fig. S5 that NEK1/C21ORF2 interaction happens in other human cell lines, this

experiment doesn't prove that there is the same dependency on each member for the interaction. Would be important to know if upon siNEK1 or siC21ORF2 in U2OS, the complex also disassembles; Watanabe et al. 2020 iScience show, for example, that in HEKS KO of C21ORF2 reduces but does not completely destabilise NEK1 levels.

We have used siRNAs to deplete NEK1 and C21ORF2 from U2-O-S and HeLa cells. Depletion of NEK1 from U2-O-S or HeLa cells reduced C21ORF2 levels (Figs. S2A, B), as we had seen in APRE-19 cells. Intriguingly, in these cell lines (unlike ARPE-19 cells), depletion of C21ORF2 caused a slight decrease in NEK1 protein levels although the size effect was variable (Figs. S2A, B). We mention that similar data were reported in HEK293 cells (Watanabe *et al*, 2020).

Alternatively, look at S-527 RAD54 or RAD51 foci in U2OS upon NEK1 siRNA depletion. Maybe a limitation of the study section could be added to address some of the limitations (another example is the use of only 1 NEK1 KO clone).

The Reviewer makes an important point – the HR reporter data we showed are from reporter cassettes integrated in U2-O-S cells, and the RAD51 data were from ARPE-19 cells. We examined, as suggested, RAD51 dynamics in IR-treated U2-O-S cells depleted of NEK1 or C21ORF2. We did four biological replicates and used several siRNAs to deplete each partner proteins. These data show that, unexpectedly, depleting NEK1 or C21ORF2 from U2-O-S cells dramatically delays RAD51 focus formation (see Extra Figure 2, below), which could in principle explain the defect seen with the HR reporter cassettes. However, the data are at odds with our data in ARPE-19 where RAD51 dynamics were unaffected by NEK1 or C21ORF2 knockout. The discrepancy may be down to differences between cell lines, transformed versus untransformed, or knockout versus siRNA. To add to the confusion, the report from the Lobrich lab showed that in HeLa cells, RAD51 loading was normal but unloading was delayed. Therefore, it appears that much more work needs to be done before making conclusions about the impact of NEK1 and C21ORF2 on RAD51 dynamics. For this reason, we have removed the ARPE-19 RAD51 data from the paper because they are potentially misleading, but we are open to discussion on this point. The main point of the figure 6 is the demonstration that depleting C21ORF2 inhibits HR in reporter cassettes similar to depleting NEK1. In revised Fig. 6B, we have strengthened these data by using 8 different C21ORF2 siRNAs; 6 of the 8 cause a substantial inhibition of HR.

6. Extreme defect on cilia formation; from the images, it seems that pericentrin is not affected, but the ARL13B is disassembled (green dispersed dots can be observed in NEK1 KO but not parental). Is NEK1 interacting with ARL13B or affecting its Y406 phosphorylation; I note that NEK1 is a mainly T-kinase but in certain conformations (see Bert van de Kooij et al. eLife. 2019; doi: 10.7554/eLife.44635; a paper that I feel should be cited in the introduction at least) its kinase activity can shift to Y ("APE-4 isoleucine in Nek1 enhances substrate phosphorylation on tyrosine residues"; the mutation is NEK1 P168I).

It was reported in 1992 by the Pawson lab that the isolated kinase domain from murine NEK1 expressed in bacteria can phosphorylate generic phospho-acceptor substrates (casein/H1/poly-Glu/Tyr) on tyrosine residues (PMID 1382974; now cited in the Intro). It's not yet clear of this activity is due to mis-folding of the recombinant protein, however. The report from van de Kooij *et al* (2019; now cited in the Intro) shows that in contrast, recombinant human NEK1 cannot phosphorylate synthetic peptides on Tyr residues but can be made to do so by mutating certain residue in the active site (now cited). On balance it seems unlikely, but not impossible, that NEK1 phosphorylates its targets on tyrosine residues in cells but it remains to be determined. We have carried out two separate phosphoproteomic screens and none of the NEK1-dependent phosphorylation events we observed occur on tyrosine residues (unpublished data). We were intrigued by the suggestion that NEK1 might phosphorylate ARL13B at residue Tyr406. We tried hard to find reference(s) describing tyrosine phosphorylation of ARL13B residue 406 but we could not find any reports or

supporting data to guide experiments. We did, however, test as suggested if NEK1 interacts with ARL13B, but we failed to detect an interaction at least in co-immunoprecipitation experiments (Extra Fig. 1 below).

Reviewer #2

Major points

1) Authors decided binding domains of NEK1 and C21ORF2, predicted more detail using AlphaFold, and validated using mutated protein. Unfortunately, it is hard to say that this information opens a new incite about biological or pathological function or NEK1 and C21ORF2.

Given that to date there is no information on the mode of interaction of NEK1 with C21ORF2, we feel that the modelling provides a preliminary framework upon which to understand it. The modelling strongly suggests that two regions in C21ORF2 interact with NEK1 like a clamp, and this mode of interaction was not apparent from other experiments. The highest-confidence interface looks like a charge zipper – a row of negative charges on NEK1 interacting with a series of positive charges in C21ORF2 also underscores the multi-valent nature of the interaction, which we believe to be an interesting feature of the complex.

2) Authors described suppression of NEK1 expression decreased C21ORF2 expression. They tried some reagents for regulation of protein stability but failed to elucidate that mechanism. To clarify the function of NEK1 or C21ORF2, authors should find suggestive mechanisms, also, should show the effect of this decreased expression on phenotype of cells. It is very understandable if there are data showing whether the effect of NEK1 KO on HR and ciliogenesis is rescued or not by C21ORF2 overexpression.

We showed in the paper that inhibiting the proteasome, autophagy or lysosomal function did not prevent the decrease in C21ORF2 levels seen in NEK1-KO cells, and therefore the mechanism is unclear. We feel that it would take longer than the time allocated for revision to tackle this problem, especially as it's not clear which direction to take.

We interpret the Reviewer's comment to enquire whether it's possible that it's the reduction in C21ORF2 levels in NEK1-KO cells could be solely responsible for the defects seen in those cells. Testing whether overexpression of C21ORF2 rescues the HR or ciliogenesis defects is difficult technically and we have not been able to test this possibility satisfactorily. However, the data in Figures 5D&E show that the kinase-dead NEK1 D146A mutant is unable to rescue the ciliogenesis defects in the NEK1 KO cells, but it fully restores C21ORF2 protein to wild-type levels. These data argue that the reduction in C21ORF2 levels cannot be responsible for the defects seen in NEK1-defective cells.

3) In this article, authors described the functions of NEK1 and C21ORF2 were observed during HR and ciliogenesis. If so, where do NEK1 and C21ORF2 exist?

This is an important point. In terms of the HR function, it is puzzling that in immunofluorescence experiments, there appears to be very little NEK1 or C21ORF2 in the nucleus (unpublished data), and this observation has been reported by others. However, cell fractionation experiments we have carried out showed a small proportion of NEK1 is in the nucleus (unpublished data), which is presumably sufficient to facilitate HR. In terms of the ciliogenesis role, in new experiments shown in revised Fig 5 (A-C), we show that NEK1 and C21ORF2 co-localize at centrosomes and at the base of primary cilia, so they are well positioned to execute control of ciliogenesis.

4) Author used cells derived from ARPE-19 cells. This cell line is derived from retinal pigment epithelia (RPE) cell, and frequently is used to observe ciliogenesis. But as authors described some pathogenesis of NEK1 and C21ORF2 is related to neuron. Whether what authors observed here also observed in neuron?

We chose ARPE-19 cells because it's an untransformed cell line. We hypothesize that the functions of NEK1 and C21ORF2 determined in non-neuronal cell lines will also apply in neurons; this will be the next phase of our study, we have started genome editing in iPSC cells to facilitate these experiments and to make conditional NEK1 knockouts in neuronal cell lines.

Minor points

5) In abstract, authors do not describe HR as well as ciliogenesis, but this review supposed relationship between binding affinity and phenotype is quite important points of this articles.

We have rectified this oversight in the revised manuscript.

6) In section of introduction author focus on topics related to these results. Author just described review of these area.

We have included more references to primary research papers in the relevant sections of the Introduction.

Reviewer #3

...The final piece of data showing that ciliogenesis is not rescued by add-back of a NEK1 mutant (D1277A) or C21ORF2 mutant (L224P) that are defective in binding the respective partner does begin to address the importance of complex formation. However, more extensive work that is beyond the scope of the current manuscript is required to rule out the possibility that these mutations act through other mechanisms and to test in detail how these two proteins regulate each other....

...Overall, I found the manuscript well written and the data to be of very high quality. However, I feel that they need to tone down some of their conclusions about the function of the NEK1:C21ORF2 complex and provide more clarity throughout the manuscript about the previously published work, explaining where their results confirm, complement or differ from previous findings. I also feel the title is not appropriate as it implies that interaction with C21ORF2 controls the cellular functions of NEK1, when the current manuscript falls some way short of showing this...

This is a fair point, which was echoed by one of the other Reviewers. We agree that more work is needed to be certain that failure of the D1277A and L224P mutations to rescue the ciliary defects of the KO cells is due to defective interaction with the partner protein. We have replaced the word “interaction” with “association” in places, and we have toned down our conclusions about the function of the complex as requested. In the first paragraph of the Discussion, we address how we might address the issue in the future.

Evidence is provided here that the endogenous NEK1 and C21ORF2 form a complex in ARPE-19 cells, although in contrast to the previous report that described a third component of the complex, SPATA7, the authors argue that, at least in ARPE-19 cells, this complex is mostly devoid of other proteins.

The study that identified SPATA7 involved overexpressed proteins whereas we immunoprecipitated the endogenous proteins from WT versus KO cells; this may explain the difference.

Overall, I found the manuscript well written and the data to be of very high quality. However, I feel that they need to tone down some of their conclusions about the function of the NEK1:C21ORF2 complex and provide more clarity throughout the manuscript about the previously published work, explaining where their results confirm, complement or differ from previous findings.

We have tried to address this point in the Discussion, refraining from explicit statements that the interaction of the two proteins controls NEK1 function, and we have toned down our conclusions. We have cited all relevant previously published work, and we have put extra information in the Intro and Discussion. One thing that didn't work so well was “explaining where their results confirm, complement or differ from previous findings”: this seemed a little contrived when we wrote the discussion with “compare/contrast” statements. However, relevant information regarding previously published papers is now provided.

I also feel the title is not appropriate as it implies that interaction with C21ORF2 controls the cellular functions of NEK1, when the current manuscript falls some way short of showing this.

This is a fair point. We have changed the title to: “Functional characterization of C21ORF2 association with the NEK1 kinase”.

There are a series of other corrections that are required as listed below.

- The Introduction, on page 4, states that "Most ALS-associated NEK1 mutations are heterozygous and are presumed to be dominant and loss-of-function in nature". This needs some explanation as most loss-of-function mutations act in a recessive manner.

We have inserted an extra sentence to provide more information.

- On page 7, the text relating to Fig. 1B should indicate the percentage of Triton X-100 used in the wash IP buffers, alongside the concentration of NaCl.

We have amended the text accordingly.

- The text talks in a couple of places about "resting, asynchronous cells"; it's not clear what it meant by "resting" and I suggest this is revised.

We have removed the word "resting" or replaced it with "unperturbed".

- On page 8, the text describing Fig 3C states "However, the S1036* and D1277A NEK1 mutants showed a dramatic reduction in their interaction with C21ORF2, and the reverse co-immunoprecipitation experiment confirmed these findings (Fig. 3C)." I couldn't find the reverse Co-IP described here?

This was a typo as we had not shown the reverse co-ip; we have removed the sentence in question.

- On page 9, there is a typo at the end of the section: L224P (text says L22P).

We have amended the text accordingly.

- It's not clear why the text discussing the AlphaFold model of the NEK1-CID indicates this to be residues 1216-1282 while Fig. 4A shows residues 1208-1286.

This was an error on our part, we have corrected the text.

- On page 10 and in Fig. 4D, the C21ORF2 combined mutant is sometimes referred to as the 3KE mutant and sometimes as the 3KRE mutant. I suggest it be consistently called the 3KRE mutant.

We have amended the text accordingly.

- On page 11, there is reference to Fig. 5C in one place when I think it should be 5E?

The figures have been renumbered and look OK now.

- On page 12, it states "However, the C21ORF2 interaction defective NEK1 mutants did not rescue NEK1 expression" when I think it should say "did not rescue C21ORF2 expression".

We have amended the text (on page 10) accordingly.

- The last section on page 12 should also describe the results with the C21ORF2-L224P mutant.

We have amended the text accordingly.

- The Discussion needs revision taking account of the points made above, including drawing more circumspect conclusions about the function of the NEK1:C21ORF2 complex.

We have removed explicit statements to the effect that the interaction of NEK1 with C21ORF2 controls X/Y/Z.

- The electrophoretic mobility shift detected for C21ORF2 when co-expressed with NEK1 in Fig. 3C is briefly mentioned in the Discussion but should also be mentioned in the Results as this is an important observation.

We have amended the Results and Discussion accordingly.

- On Figure 5C and E, it appears that the assay used is incorrectly labelled as I believe Fig. 5C is the TLR assay and Fig. 5E is the DR-GFP assay.

This figure has been revamped, with extra data added and the schematic diagrams moved to Fig. S8.

- Fig. S1 and S6A differ in what is labelled as the 'acidic region' of NEK1. This should be consistent. The legend to Fig. S1 should also explain the orange and purple boxes in NEK1, and the blue and green boxes in C21ORF2.

We have corrected this.

[Figures removed by LSA editorial staff per authors' request]

April 11, 2023

RE: Life Science Alliance Manuscript #LSA-2022-01740R

Prof. John W Rouse
University of Dundee
MRC Protein Phosphorylation and Ubiquitylation Unit
Hawkhill
Dundee DD1 5EH
United Kingdom

Dear Dr. Rouse,

Thank you for submitting your revised manuscript entitled "Functional characterization of C21ORF2 association with the NEK1 kinase". We would be happy to publish your paper in Life Science Alliance pending final revisions necessary to meet our formatting guidelines.

- please address Reviewer 2's comments about the Methods section
- please upload your supplementary figure files as single files and add your supplementary figure legends to the main manuscript text
- please add the Twitter handle of your host institute/organization as well as your own or/and one of the authors in our system
- please make sure that the author list in the manuscript and our system match and that each author is listed in our system

Figure Check:

- please add scale bars to Figure 5D

A. FINAL FILES:

B. MANUSCRIPT ORGANIZATION AND FORMATTING:

Sincerely,

Reviewer #1 (Comments to the Authors (Required)):

I am delighted that the authors have responded positively to my comments and improved their manuscript significantly. Their response indicates that they have carefully considered my suggestions and made the necessary changes to address them.

I appreciate that the authors have reorganized the paper, eliminated unnecessary repetition, rectified the typos, and provided data files, as suggested. They have also responded to my concerns regarding the RAD51 foci, HR defect without RAD51/RAD54 data, and cilia formation, providing additional data and explanations.

Based on the authors' response, I believe that the manuscript is now in a publishable state.

Reviewer #2 (Comments to the Authors (Required)):

The revised manuscript is well-written, and gets easily understandable for readers. However, unfortunately there are no mechanisms or functional relationship between NEK1 and C21ORF2, just show colocalization of these proteins.

For HR study in Figure 6B, authors tried to C21ORF2 knock down using 6 different oligos. I appreciate the authors' efforts, but they did not show knock-down efficiency of these oligos. If the effects on HR are parallel of knock-down efficiency, the data will be compelling.

It is quite difficult to follow their experimental procedures, because the section of methods was not well organized. For example, there are two paragraphs for immunoprecipitation. Authors did not describe what kind of beads they used, etc.

April 25, 2023

RE: Life Science Alliance Manuscript #LSA-2022-01740RR

Prof. John W Rouse
University of Dundee
MRC Protein Phosphorylation and Ubiquitylation Unit
Hawkhill
Dundee DD1 5EH
United Kingdom

Dear Dr. Rouse,

Thank you for submitting your Research Article entitled "Functional characterization of C21ORF2 association with the NEK1 kinase mutated in human in disease". It is a pleasure to let you know that your manuscript is now accepted for publication in Life Science Alliance. Congratulations on this interesting work.

DISTRIBUTION OF MATERIALS:

Again, congratulations on a very nice paper. I hope you found the review process to be constructive and are pleased with how the manuscript was handled editorially. We look forward to future exciting submissions from your lab.

Sincerely,
